# SARS-CoV-2-ORF-3a Mediates Apoptosis Through Mitochondrial Dysfunction Modulated by the K^+^ Ion Channel

**DOI:** 10.3390/ijms26041575

**Published:** 2025-02-13

**Authors:** Muhammad Suhaib Qudus, Uzair Afaq, Siyu Liu, Kailang Wu, Chen Yu, Mingfu Tian, Jianguo Wu

**Affiliations:** 1State Key Laboratory of Virology, College of Life Sciences, Wuhan University, Wuhan 430072, China or suhaibq121@gmail.com (M.S.Q.); 2018172040001@whu.edu.cn (U.A.); 2020202040047@whu.edu.cn (S.L.); wukailang@whu.edu.cn (K.W.); jwu@whu.edu.cn (J.W.); 2Key Laboratory of Ministry of Education for Viral Pathogenesis & Infection Prevention and Control, Institute of Medical Microbiology, Jinan University, Guangzhou 510632, China

**Keywords:** *SARS-CoV-2*, ORF-3a, apoptosis, mitochondrial dysfunction, Mitochondrial ATP-sensitive Potassium Channel (MitoK^ATP^)

## Abstract

Coronavirus disease 2019 (COVID-19) causes pulmonary edema, which disrupts the lung alveoli–capillary barrier and leads to pulmonary cell apoptosis, the main cause of death. However, the molecular mechanism behind *SARS-CoV-2*’s apoptotic activity remains unknown. Here, we revealed that SARS-CoV-2-ORF-3a mediates the pulmonary pathology associated with *SARS-CoV-2*, which is demonstrated by the fact that it causes lung tissue damage. The in vitro results showed that SARS-CoV-2-ORF-3a triggers cell death via the disruption of mitochondrial homeostasis, which is modulated through the regulation of Mitochondrial ATP-sensitive Potassium Channel (MitoK^ATP^). The addition of exogenous Potassium (K^+^) in the form of potassium chloride (KCl) attenuated mitochondrial apoptosis along with the inflammatory interferon response (IFN-β) triggered by SARS-ORF-3a. The addition of exogenous K^+^ strongly suggests that dysregulation of K^+^ ion channel function is the central mechanism underlying the mitochondrial dysfunction and stress response induced by SARS-CoV-2-ORF-3a. Our results designate that targeting the potassium channel or its interactions with ORF-3a may represent a promising therapeutic strategy to mitigate the damaging effects of infection with *SARS-CoV-2*.

## 1. Introduction

The emergence of *severe acute respiratory syndrome coronavirus 2* (*SARS-CoV-2*) infection, resulting in coronavirus disease 2019 (COVID-19), has had an immense worldwide impact, with over 7 million deaths (https://covid19.who.int) [1]. Severe disease is more common in vulnerable groups, such as the old, young, immunocompromised, overweight, and those with pre-existing co-occurring conditions including diabetes and obesity [2]. *SARS-CoV-2* infections are highly contagious and possibly lethal, with both asymptomatic carriers and symptomatic patients experiencing fatigue, body pains, dry cough, and fever. In severe situations, these symptoms may progress to viral pneumonia, dyspnea, multi-organ failure, and eventually death [3,4]. Asymptomatic *SARS-CoV-2* carriers develop a defense mechanism via balanced antiviral cellular immunity [5].

Apoptosis, a type of modified cell demise, is critical in antiviral safeguarding, empowering the host to restrict viral spread by killing contaminated cells and lessening resistant overactivation [6]. In contrast to other cell death processes such as necroptosis and pyroptosis, apoptosis is evolutionarily conserved and critical for viral infections because it precisely regulates immune responses [7]. Recent evidence suggests that coronaviruses, such as *SARS-CoV* and *MERS-CoV*, induce apoptosis, contributing to organ damage and disease progression [8,9]. Clinical studies of COVID-19 patients report apoptosis in endothelial cells of lung tissue, with *SARS-CoV-2* inducing apoptosis in alveolar cells in human lung stem cell-derived alveolospheres [10,11]. Similarly, Syrian hamsters and humanized *ACE2* transgenic mice have shown substantial cell death in the lung epithelium, confirming *SARS-CoV-2*’s capacity to trigger apoptosis in vivo [12,13,14].

The interplay of cellular and viral proteins can control apoptosis [8]. Like other coronaviruses, *SARS-CoV-2* consists of four structural proteins, membrane (M), envelope (E), nucleocapsid (N), and spike (S), as well as non-structural proteins and potential accessory components, including ORF-3a [15,16,17,18]. ORF-3a, the largest accessory protein of SARS-CoV-2, belongs to the viroporin family, shares 72.7% identity with *SARS-CoV*, and consists of 275 amino acid sequences [16,19]. Although the *SARS-CoV* and *SARS-CoV-2* ORF3a proteins are similar, their structural and functional distinctions set them apart as distinct entities, with SARS-CoV-2-ORF3a displaying distinctive biological roles [20]. The *SARS-CoV* viroporins, including M, E, 7a, and ORF-3a, are necessary for viral replication and pathogenicity [21,22]. It has previously been proposed that the accessory protein ORF-3a, encoded by *SARS-CoV*, causes cell death [23,24]. However, additional research is needed to identify the potential role of ORF-3a proteins and molecular mechanisms contributing to SARS-CoV-2-induced apoptosis.

Mitochondria are a prominent target in *SARS-CoV-2* infection, serving an important part in apoptotic signaling. According to proteomic analyses of host cells infected with *SARS-CoV-2*, infection increases the levels of mitochondrial apoptotic mediators, including cytochrome c (cyt-C), a crucial pro-apoptotic protein [25]. Mitochondrial apoptosis is regulated by the B-cell lymphoma-2 (BCL-2) family, which includes pro- and anti-apoptotic proteins. Pro-apoptotic proteins, such as BOK (BCL-2 ovarian killer protein)*,* go to the mitochondria and release cyt-C to start apoptosis, which signals the death of cells downstream [26].

*SARS-CoV-2-ORF-3a* triggered erythrocyte apoptosis by forming non-selective calcium (Ca^2+^) permeable cation channels, an adaptation of the novel dimeric shape that is highly conserved among *Coronaviridae*. Recent results show that SARS-CoV-2-ORF-3a is localized in the endoplasmic reticulum (ER), where it induces reticulophagy regulator 1 (RETREG1)-dependent reticulophagy via the high mobility group box1- beclin1 (HMGB1-BECN1) pathway, causing ER stress. As a result, it can cause cell death [27]. However, the mechanism of mitochondrial apoptosis induction by SARS-CoV-2-ORF-3a through the potassium ion (K^+^) ion channel remains unclear.

Here, we inspect the role of SARS-CoV-2-ORF-3a in inducing apoptosis and mitochondrial dysfunction via K^+^ ion channels. Our study highlights the importance of the K^+^ ion channel’s involvement in SARS-CoV-2-ORF-3a-induced mitochondrial apoptosis and inflammatory response. By altering the K^+^ ion gradient, pharmacological interventions targeting these channels may inhibit excessive apoptosis, inflammation, and subsequent lung injury, alleviating the severe respiratory complications associated with COVID-19. In the context of *SARS-CoV-2* infection, it highlights the therapeutic potential of targeting potassium ion channels, providing a promising route for future research and clinical applications.

## 2. Results

### 2.1. SARS-CoV-2 Variants Induce Lung Tissue Damage

The tissue damage caused by *SARS-CoV-2* is complex and can affect multiple organ systems [28]. To investigate *SARS-CoV-2*-induced apoptosis in lung tissue, we infected C57BL/6 mice with *SARS-CoV-2 virus wild type* (100 pfu and 1000 pfu), *Delta* (1000 pfu), and *Omicron* (*BA.2 strain*, 1000 pfu). We performed hematoxylin and eosin (H&E) staining and the terminal deoxynucleotidyl transferase dUTP nick end labeling (TUNEL) assay. The H&E picture revealed abnormal cell morphology, nuclear abnormality, eosinophilia, hemorrhage, and increased alveolar space signifying pulmonary edema, inflammation, and tissue injury compared to the control (mock) (Figure 1a,b). The TUNEL staining demonstrated an increased number of TUNEL-positive cells in infected C57BL/6 mice compared to control mice (Figure 1c,d). In *SARS-CoV-2* pathogenesis, the host immune response is pivotal, particularly in recruiting and activating inflammatory cells [29]. To study the immune response, we performed immunohistochemistry (IHC) of the lungs infected with the *SARS-CoV-2* strains mentioned above for immune response. The IHC analysis revealed distinct immune cell infiltration patterns in mice lungs infected with *SARS-CoV-2* strains. Compared to the mock group, the wild-type *SARS-CoV-2* infection, mice infected with the *Delta* and *Omicron BA.2* variants resulted in a significant increase in F4/80-positive macrophage infiltration in the lung tissue. The *Delta variant* group exhibited the highest number of F4/80-positive cells, followed by the *wild-type* and *Omicron BA.2 variant* (Figure 1e,f). Similarly, Ly6G-positive neutrophil infiltration was assessed. Mice infected with the above-mentioned variants demonstrated a remarkable increase in Ly6G-positive neutrophil infiltration in the lung tissue compared to the mock group. The *Delta variant* showed the most prominent neutrophil accumulation (Figure 1g,h). These results highlight the distinct patterns of lung pathology and innate immune cell recruitment elicited by different *SARS-CoV-2 variants* in mice models. The *wild-type* and *Delta variants* appear to induce a more severe lung injury and a robust inflammatory response. In contrast, the *Omicron variant* is associated with relatively milder histological changes and dampened immune cell infiltration. The mitochondrial ion channels found in the IMM (inner mitochondrial membrane) and the OMM (outer mitochondrial membrane) are widely acknowledged as key actors in controlling mitochondrial function [30]. The potassium ion channel plays a crucial role in maintaining mitochondrial hemostasis. CCDC51 is known as MitoK^ATP^, essential in regulating mitochondrial function and apoptosis. The activation of CCDC51 (MitoK^ATP^) channels has been implicated in various physiological and pathological processes, including apoptosis induced by cellular stress [31,32]. In addition to immune cell infiltration, we assessed the role of the mitochondrial K^ATP^ (MitoK^ATP^) channel which is localized to the inner mitochondrial membrane through IHC in the lungs of mice infected with *SARS-CoV-2 variants*. The IHC analysis showed a significantly higher MitoK^ATP^ (brown punctate) expression than the mock group, indicating increased mitochondrial stress. The *wild-type* and *Delta variants* showed a more pronounced expression of MitoK^ATP^ (Figure 1i,j), in contrast to the *Omicron variant*. The differential expression of MitoK^ATP^ in response to these variants suggests that the mitochondrial ion channel may play a role in the severity of the infection. The observed disparities in disease severity and clinical outcomes may be attributed to these variant-specific differences in the host response.

### 2.2. SARS-CoV-2-ORF-3a Induces Apoptosis

Despite identifying apoptosis pathways implicated in the infection with *SARS-CoV-2*, the specific mechanism by which the virus controls cell apoptosis (programmed cell death) is still largely unknown [33,34,35,36]. To investigate whether *SARS-CoV-2* induces cell apoptosis, we first expressed different viral proteins *(ORF-3a*, *E*, *M*, *N*, *6*, and *8)* of *SARS-CoV-2* into HeLa cells upon transfection by Western blotting (Figure 2a). To evaluate the apoptotic activity of ORF-3a alone, we transfected HeLa cells with ORF-3a along with an empty vector (Flag-Vec) and a mock control. Western blot analysis for cleaved caspase-3 demonstrated that ORF-3a alone induces apoptosis compared to Flag-Vec and the mock control (Figure 2b). Although ORF-3a alone can initiate apoptosis, additional apoptotic treatments such as cisplatin (CCDP), carbonyl cyanide m-chlorophenyl hydrazone (CCCP), Doxorubicin (Dox), and vesicular stomatitis virus (VSV) were used to investigate how ORF-3a interacts with different apoptotic pathways and whether it sensitizes the cells to other forms of apoptosis. We subsequently evaluated whether other *SARS-CoV-2* proteins, in addition to ORF-3a, could induce apoptosis upon treatment with CCDP for 6 h. Among the tested proteins, ORf-3a significantly increased the induction of cleaved caspase-3 (Figure 2c), an important executioner during apoptosis. Next, we transfected HeLa cells with ORF-3a and Flag-Vec and treated them with CCDP for 6 h. Western blotting results confirmed that SARS-CoV-2-ORF-3a promotes cell apoptosis (Figure 2d). Further, we conducted quantitative analysis through flow cytometry using Annexin V-FITC/PI staining as per the manufacturer’s procedure, confirming that SARS-CoV-2-ORF-3a triggers apoptosis in HeLa cells supplemented with CCDP in comparison to an empty vector (Figure 2e,f). Extending our investigations into apoptotic effects of SARS-CoV-2-ORF-3a, we applied a series of treatments such as CCCP, Dox, and VSV. Following the results obtained with CCDP, CCCP treatment for 30 min elicited a pronounced increase in apoptotic biomarkers as evidenced by Western blot analysis. Specifically, Western blot bands showed an enhancement of cleaved caspase-3, indicative of the initiation of apoptosis. Flow cytometric analysis with Annexin V-FITC/PI further substantiated the induction of apoptosis, revealing an increase in apoptotic cell population post-CCCP treatment (Figure 2g–i). Similarly, Dox (24 h) was used to mimic the above experimental settings, resulting in the elevated expression of the apoptotic marker (cleaved caspase-3) in HeLa cells that expressed ORF-3a. The apoptotic markers were upregulated, as shown in prior treatments, according to Western blot analysis (Figure 2j). Quantitative flow cytometry demonstrated a substantial rise in the frequency of apoptotic cells when treated with Dox (Figure 2k,l). VSV is a perfect choice for studying viral-induced apoptosis due to its strong cytopathic effects. Following ORF-3a transfection, HeLa cells were infected with VSV (20 μL, MOI =1) for 12 h, and Western blot analysis resulted in an upregulation of the apoptotic marker (Figure 2m). Apoptosis was validated by flow cytometry, which showed that after VSV infection, more ORF-3a-transfected cells were stained for Annexin V-FITC/PI (Figure 2n,o). The introduction of CCDP, CCCP, Dox, and VSV strengthened our findings that ORF-3a-induced apoptosis is not limited to a single apoptotic pathway but rather potentiates apoptosis through mechanisms activated by external stressors such as DNA damage and oxidative stress.

### 2.3. SARS-CoV-2-ORF-3a Induces Apoptosis Through Mitochondrial Dysfunction

The apoptotic pathway may be centered on the mitochondrial malfunction, which induces apoptosis. ROS production in animal cells is primarily attributed to the mitochondrial respiratory chain and mitochondrial lipid peroxidation. Extreme reactive oxygen species (ROS) production causes oxidative stress and dysregulation of energy metabolism [4]. To inspect the potential role of the SARS-CoV-2-ORF-3a protein in inducing oxidative stress, we transfected HeLa cells with SARS-CoV-2-ORF-3a for 24 h and analyzed ROS production using fluorescence microscopy and flow cytometry. The surge in green fluorescence in HeLa cells with SARS-CoV-2-ORF-3a compared to the Flag-vector control demonstrated elevated ROS production (Figure 3a). Quantitative analysis through flow cytometry revealed a significant rise in ROS levels in HeLa cells transfected with SARS-CoV-2-ORF-3a (Figure 3b,c). Although the exact localization of ROS was not directly assessed, the observed increase in ROS production suggests that ORF-3a could potentially interfere with mitochondrial function, resulting in heightened ROS production and an overall rise in oxidative stress levels.

Depolarization of the transmembrane potential (Δψm), release of apoptogenic substances, and loss of oxidative phosphorylation can be induced by opening the mitochondrial permeability transition pore [37]. We investigated the impact of SARS-CoV-2-ORF-3a on mitochondrial function by staining HeLa cells with JC-1 dye to identify Δψm. In healthy mitochondria, the JC-1 dye accumulates and fluoresces red, while upon loss of Δψm or depolarized mitochondria, it appears as monomers and fluoresces green. We found that mitochondrial membrane potential is disrupted by the reduction in the intensity of red fluorescence as well as the surge in green fluorescence (monomers), upon transfection of SARS-CoV-ORF-3a into HeLa cells (Figure 3d). Δψm alteration was quantified through flow cytometry analysis on HeLa cells stained with JC-1 after transfection with SARS-CoV-2-ORF-3a. The ratio of red fluorescence (JC-1 aggregates) to green fluorescence served as the metric for Δψm. A reduced red/green fluorescence ratio signifies a greater decrease in Δψm (Figure 3e,f).

Apoptotic components released from the mitochondrial intermembrane space into the cytosol are caused by the opening of the mitochondrial permeability transition pore (MPTP), which occurs when Δψm is lost. SARS-CoV-2-ORF-3 was transfected into HeLa cells and treated with apoptotic inducers, CCDP, and VSV. The mitochondrial and cytoplasmic fractions were isolated from HeLa cells, and the release of cytochrome c (cyt-C), a key pro-apoptotic factor, was evaluated by Western blotting. A significant increase in cyt-C levels was observed in the cytosolic fraction, indicating its release from the mitochondria, which is a hallmark of apoptotic signaling. This release is often associated with mitochondrial outer membrane permeabilization (MOMP), a critical event in apoptotic signaling. In addition to cyt-C, the expression of Tom-20, a marker of mitochondrial membrane integrity and a component of the translocase of the outer mitochondrial membrane (TOM complex), was also assessed. Interestingly, the increase in Tom-20 within the mitochondrial fraction could indicate alterations in mitochondrial membrane integrity or the dynamics of the mitochondrial membrane protein complex during apoptotic processes. (Figure 3g,h). These findings highlight that cyt-C release into the cytosol and the increased expression of Tom-20 within the mitochondria are both indicative of mitochondrial dysfunction during apoptosis. This disruption of mitochondrial membrane potential results in the release of pro-apoptotic factors which in turn may enable the efflux of mt-RNA into the cytosol.

Additionally, the release of mitochondrial RNA (mt-RNA) into the cytosol was observed, further supporting mitochondrial dysfunction. The leakage of mt-RNA indicates compromised mitochondrial membrane integrity, allowing mitochondrial genetic material to escape into the cytoplasm. To further investigate the impact of mitochondrial dysfunction, qPCR was performed to quantitatively assess the expression of key mitochondrial genes: Cox-1, CYB, and ND5. These genes are crucial for mitochondrial oxidative phosphorylation and energy production.

In HeLa cells transfected with SARS-CoV-2-ORF-3a and treated with CCDP or *VSV*, the expression levels of Cox-1, CYB, and ND5 were significantly elevated compared to controls, with the expression of each gene normalized to GAPDH as an internal control. These results (Figure 3i–n) indicate that ORF-3a transfection promotes mitochondrial dysfunction, as evidenced by the increase in mitochondrial gene expression alongside other indicators of mitochondrial stress, such as cyt-C release and changes in Tom-20 expression.

Taken together, these findings suggest that ORF-3a induces mitochondrial dysfunction, characterized by the production of reactive oxygen species (ROS), loss of mitochondrial membrane potential (Δψm), opening of the mitochondrial permeability transition pore (MPTP), and the release of both cyt-C and mt-RNA into the cytosol. These markers collectively reflect the disruption of mitochondrial integrity and the activation of apoptotic signaling pathways in response to SARS-CoV-2-ORF-3a.

### 2.4. Exogenous Potassium (K^+^) Supplementation Attenuates SARS-CoV-2 ORF-3a-Induced Mitochondrial Membrane Potential (ΔΨm) Disruption

The potassium (K^+^) ion channel plays a crucial role in mitochondrial function and in maintaining mitochondrial homeostasis. These ion channels regulate the influx and efflux of potassium ions across the mitochondrial membrane. Potassium ion channels are critical in maintaining the electrochemical gradient across the mitochondrial membrane [38,39]. We hypothesized that the K^+^ ion gradient is interrupted by SARS-CoV-2-ORF-3a. To discover the outcome of the K^+^ ion channel in the modulation of mitochondrial dysfunction, we transfected SARS-CoV-2-ORF-3a into HeLa cells and treated them with exogenous K^+^ in the form of KCl (60 mM). Fluorescent microscopy reveals that the addition of exogenous K^+^ leads to a reduction in ROS generation compared to ORF-3a-transfected cells alone (Figure 4a). Flow cytometry was used to quantify ROS production. The analysis revealed a significant decrease in ROS production compared to ORF-3a-transfected cells alone (Figure 4b,c). Next, we examined the effect of the K^+^ ion channel on m Δψm, JC-1 staining was performed to detect Δψm on HeLa cells after the addition of exogenous K^+^. The fluorescent microscopy confirms a decrease in depolarization of Δψm compared to ORF-3a-transfected cells alone (Figure 4d). Flow cytometry analysis is performed to quantify the fluorescence intensity of JC-1 dye in transfected cells. Flow cytometry showed that the green fluorescence (JC-1 monomers) was decreased and the red fluorescence (Jc-1 aggregates) was increased. These alterations showed that after treatment with exogenous K^+^, mitochondrial depolarization is decreased (Figure 4e,f). Furthermore, we analyzed the effect of the K^+^ ion channel on the release of cytochrome c. We treated ORF-3a-transfected HeLa cells with CCDP, VSV, and CCCP, after adding exogenous K^+^, and then we performed Western blotting. The results presented a decrease in the expression of cytochrome c (Figure 4g–i). These results indicated that SARS-CoV-2-ORF-3a, upon the addition of l, influences ROS production, Δψm, and the release of cytochrome c. Adding exogenous K^+^ effectively boosted extracellular K^+^ concentration, reducing ROS generation, Δψm, and cytochrome c release. The potassium ion channel regulated by ORF-3a may modulate mitochondrial activity and cellular redox balance, potentially affecting cellular functions including apoptosis.

### 2.5. Exogenous K^+^ Attenuates SARS-CoV-2-ORF-3a-Induced Inflammatory Response and Apoptosis

IFN-β is a critical component of the innate immune response against viral infections [40]. To comprehend how the host antiviral response and the SARS-CoV-ORF-3a protein interact, we examine whether SARS-CoV-2-ORF-3a induces the expression of IFN-β, and we transfected an empty vector (Flag-Vec) and ORF-3a alone into HeLa cells; the results of qPCR analysis revealed that ORF-3a alone upregulated the IFN-β expression compared to the control, suggesting that ORF-3a can activate the innate immune response (Figure 5a). This was further confirmed by the IFN-β expression level in ORF-3a-transfected cells treated with different apoptotic inducers, i.e., CCDP, CCCP, and VSV. The mRNA level determined by qPCR analysis revealed a substantial surge in the expression of IFN-β as compared to the control (Figure 5b–d).

To further inquire about the interplay between ion channels and the induction of inflammatory responses, we first assessed the impact of exogenous K^+^ supplementation on IFN-β expression in HeLa cells transfected with Flag-vec and ORF-3a alone. Upon treatment with KCl, we observed a significant reduction in IFN-β expression in ORF-3a-transfected cells, suggesting that potassium ions influence the baseline inflammatory response induced by SARS-CoV-2-ORF-3a (Figure 5e). Next, we expanded our investigation by exposing ORF-3a-transfected HeLa cells to apoptotic inducers, including CCDP, CCCP, and VSV. Following these treatments, the addition of KCl led to the suppression of IFN-β expression, indicating that potassium ion channels modulate the inflammatory response even under conditions of apoptotic stress (Figure 5f–h). These findings suggest that potassium supplementation may attenuate the IFN-β response in both basal and stress-induced conditions, highlighting the interplay between potassium ion channels and the immune response in the context of SARS-CoV-2-ORF-3a-induced apoptosis.

Furthermore, we explore the effect of the K^+^ ion channel on apoptosis by evaluating the expression of cleaved caspase-3, a key marker of cell death. The Western blot examination demonstrated a drop in the expression of caspase-3 in the ORF-3a-transfected cells treated with KCl compared to the cells treated with CCDP, CCCP, and VSV alone. This suggests that exogenous K^+^ supplementation effectively mitigates apoptosis induced by ORF-3a, potentially through the modulation of mitochondrial and cellular homeostasis (Figure 5i–k). Collectively these results highlight the intricate relationship between IFN-β expression, potassium ion channels, and apoptosis induction. Exogenous potassium ions provided through KCl have an inhibitory effect on IFN-β, which is positively regulated by potassium ion channels after treatment with CCDP, VSV, and CCCP. These discoveries advance our knowledge of the intricate connection between potassium ion channels and the biological mechanisms underlying immune responses and death.

### 2.6. SARS-CoV-2-ORF-3a Induces Lung Pathology and Inflammatory Response

The SARS-CoV-2-ORF-3a protein is a multifunctional accessory protein. To investigate the role of the ORF-3a pathogenic mechanism, including induction of apoptosis modulation of host immune responses, we transfected SARS-CoV-2-ORF-3a into C57BL/6 mice using an Entranster in vivo DNA transfection reagent, and after 24 h, we dissected lungs from transfected mice. The expression of ORF-3a was quantified using qPCR. The mRNA level of ORF-3a was significantly increased in mice lung tissue (Figure 6a), validating the delivery and expression of SARS-CoV-2-ORF-3a compared to the mock control. Next, we performed H&E and TUNEL staining on the isolated lungs from the transfected mice with SARS-CoV-2-ORF-3a. The H&E staining of the lung section exhibited widespread inflammatory cell infiltration, alveolar edema, and disruption of the alveolar architecture in the SARS-CoV-2-ORF-3a-transfected group compared to the mock group (Figure 6b,c). The TUNEL assay demonstrated a higher number of apoptotic cells in the SARS-CoV-2-ORF-3a group compared to the mock group, suggesting that SARS-CoV-2-ORF-3a induced cell death. (Figure 6d,e). Furthermore, we examine the inflammatory response by performing IHC of the transfected mice lungs. According to the IHC analysis, the number of F4/80-positive macrophages increased significantly (Figure 6f,g) as well as the number of Ly6G-positive neutrophils (Figure 6h,i). In addition to inflammatory responses, we assess the role of the potassium ion channel MitoK^ATP^. The IHC analysis of MitoK^ATP^ revealed significant changes in mitochondrial ion channel expression. The ORF-3a-transfected mice lung tissue showed enhanced punctate staining of MitoK^ATP^ compared to the mock group, indicative of increased mitochondrial stress (Figure 6j,k). Next, we performed qPCR to analyze the interferon-beta (IFN-β) level expression in lung tissue transfected with ORF-3a of SARS-CoV-2. The mRNA level of INF-β was significantly augmented in comparison to the mock group (Figure 6l). These results shed light on the expression of the SARS-CoV-2-ORF-3a protein in the lungs of mice, leading to significant histological changes, including inflammatory cell infiltration, alveolar damage, and apoptosis induced by mitochondrial damage. The SARS-CoV-2 ORF-3a-induced lung pathology is accompanied by a pronounced recruitment of macrophages and neutrophils, as well as upregulation of MitoK^ATP^ expression, reflecting mitochondrial stress and dysfunction. Furthermore, the increased MitoK^ATP^ expression correlated with the observed inflammatory and apoptotic response. In addition, the upregulation of INF-β expression further links mitochondrial dysfunction to the activation of an innate immune response. These findings collectively highlight the critical role of mitochondrial ion channels in modulating the inflammatory and apoptotic responses in lung tissue following ORF-3a transfection, reinforcing the potential link between mitochondrial damage and immune activation in viral pathogenesis.

## 3. Discussion

Apoptotic pathways are essential physiological processes that are vital for maintaining the homeostasis of pathogenic microorganisms and organisms [41,42]. The intricate function of virus-induced apoptotic cell death in host antiviral immunity is unclear; it may aid in viral clearance or contribute to virus-induced tissue damage and the development of illness [43]. Here, we report that *SARS-CoV-2* triggers cell death (apoptosis) in lung tissue by activating caspase-3 and potassium ion channel (MitoK^ATP^). In vivo, the results displayed an important insight into the pathogenic mechanisms associated with different *SARS-CoV-2 variants* including *wild-type*, *Delta*, and *Omicron BA.2*. Our results demonstrated that intranasal infection with *wild-type*, *Delta*, and *Omicron BA.2 variants* of *SARS-CoV-2* led to significant histopathological changes in the lungs, including inflammatory cell infiltration, alveolar damage, and increased apoptosis.

Some studies have already evaluated the pathogenic potential of different *SARS-CoV-2 variants* in animal models. A study by Hou et al. found that the *Delta variants* lead to greater lung pathology, viral load, and inflammatory responses than the ancestral *SARS-CoV-2* [44]. Similarly, Abdelnabi et al. showed that the *Omicron variant* was far milder than the *Delta variant* in Syrian hamsters, which they accredited to differences in viral fitness and tropism [45]. Our results also agree with reports of pronounced inflammatory response and extensive infiltration of macrophages (f4/80) and neutrophils (LY6G) seen in *SARS-CoV-2 variant* pathogenesis. Liao et al. observed that the *Delta variant* evoked a heightened perturbation of cellular immunity through enhanced upregulation of inflammasomes in airway epithelium compared to the wild-type strain [46]. Additionally, Meng et al. suggested that the *Omicron variant* may have a higher capacity for immune evasion, potentially leading to a dampened inflammatory response [47]. In our study, the IHC targeting CCDC51 (MitoK^ATP^) revealed upregulation during intranasal infection with *SARS-CoV-2 variants*. The *wild-type* and *Delta variants* showed a surge in MitoK^ATP^ expression compared to *Omicron* and mock variants, suggesting that MitoK^ATP^ is actively involved in modulating mitochondrial dynamics during viral infection-induced apoptosis. The differential expression of CCDC51 across various *SARS-CoV-2 variants* may contribute to the observed differences in disease severity and inflammatory responses. Our findings align with the findings demonstrating that potassium ion channels influence apoptotic pathways by modulating cytochrome c release from the mitochondria—an essential step for caspase activation during apoptosis [48,49]. These findings highlight the importance of mitochondrial ion channels, particularly CCDC51, in the complex interplay between viral pathogenesis and host cellular responses in COVID-19. Collectively, our study finds that *SARS-CoV-2* induces both apoptosis and robust inflammatory response in mice lungs, unlike other respiratory viruses such as the *influenza virus* and *respiratory syncytial virus (RSV)*, which tend to either promote apoptosis while inhibiting inflammatory response and vice versa [50,51]. In line with our findings, the ORF3a protein in *SARS-CoV-2* variants plays a key role in pathogenesis and virulence, with notable differences between the *Delta* and *Omicron variants.* ORF3a contributes to viral virulence by interacting with host cell mechanisms, particularly by triggering inflammatory pathways, apoptosis, and cellular stress responses [52,53]. However, mutations within ORF3a across variants may alter these effects. Studies show that *Delta retains* mutations that enhance ORF3a’s ability to activate inflammatory responses, whereas Omicron has mutations that may reduce ORF3a-induced apoptosis and immune activation [21,54]. These mutations may contribute to the relatively lower cytotoxicity observed in *Omicron* than in *Delta*, aligning with the lower severity and tissue damage seen with *Omicron* infections [55,56].

*SARS-CoV-2* proteins have been shown to play a noteworthy role in triggering apoptosis in several ways, which are similar to previous findings of SARS-CoV-ORF-3a [57,58]. Our findings demonstrate that ORF-3a alone can trigger caspase-3 activation and apoptosis in HeLa cells, highlighting its pro-apoptotic capability. This study further confirms that SARS-CoV-2-ORF-3a not only initiates caspase-3-dependent apoptosis independently but also sensitizes HeLa cells to additional apoptotic stimuli, i.e., CCDP, CCCP, Dox, and VSV. The synergistic effects observed with these inducers underscore the multifaceted role of ORF-3a in modulating apoptosis pathways. These results align with the induction of pro-apoptotic activities previously described for these stimulators, suggesting that ORF-3a may serve as a crucial mediator in the apoptotic response elicited by SARS-CoV-2 [59,60,61,62]. Thus, the ability of ORF-3a to enhance apoptosis through various mechanisms emphasizes its potential role in the pathogenesis of COVID-19.

ROS are produced metabolically by mitochondria as a part of the respiratory electron transport chain. Normally, the ROS generation and the antioxidant system are in a delicate balance. On the other hand, ROS overproduction can upset this equilibrium, trigger oxidative stress, and cause cell death [63]. Intracellular ROS protect cell signaling pathways at low concentrations and can regulate protein expression to stimulate cell growth and differentiation. On the other hand, ΔΨm could be compromised if excessive ROS are produced. This is consistent with our finding in HeLa cells transfected with SARS-CoV-2-ORF-3a. The mitochondrial membrane undergoes conformational changes after transfection of SARS-CoV-2-ORF-3a into HeLa cells, increasing ΔΨm and the subsequent release of cyt-C, which aligned with previously published research [64,65].

The mitochondrial ion channels found in the IMM (inner mitochondrial membrane) and the OMM (outer mitochondrial membrane) are widely acknowledged as key actors in controlling mitochondrial function [39,66]. The potassium ion channel is crucial for maintaining mitochondrial homeostasis by maintaining the K^+^ electrochemical gradient across the inner mitochondrial membrane, which is crucial for maintaining ΔΨm and its function [67,68]. The current study demonstrates that supplementing cells transfected with ORF-3a with extracellular KCl decreased the generation of ROS, the ensuing loss of ΔΨm, and the release of cytochrome c that ORF-3a induced. Our results are in line with previously reported studies [69,70,71].

Interferons play a crucial role in the innate immune response against viral infections [72]. Our results showed that the SARS-CoV-2-ORF3a protein alone and also when exposed to other apoptotic inducers, i.e., CCDP, CCCP, and VSV, cause an inflammatory reaction, as shown by the increase in IFN-β expression. The mitochondrial malfunction and cellular stress caused by ORF3a most likely lead to this inflammatory response. The elevation of IFN-β expression that ORF-3a has been shown to induce is consistent with earlier findings that ORF3a can stimulate innate immunity pathways, including the cGAS-STING pathway, which in turn produces type I interferons [52,73]. With the dual function of ORF-3a in promoting cell damage through apoptosis and inflammation, as well as stimulating the antiviral interferon response, any therapeutic strategy targeting ORF-3a must be approached with caution. Hindering ORF-3a could assist with lessening lung tissue damage from *SARS-CoV-2* but also risks reducing the protective IFN-β response that assumes a critical part in restricting viral replication [52,74]. To maintain the induction of interferon, efforts targeting ORF-3a should strive to balance its pro-apoptotic and inflammatory roles [75]. Further studies on the structural and functional variations in ORF-3a across *SARS-CoV-2 variants* could help develop selective inhibitors that mitigate tissue damage without compromising innate immune responses [76]. Interestingly, our findings demonstrate that ORF-3a alone is capable of activating IFN-β expression, indicating its role in initiating an antiviral response. Moreover, the addition of exogenous K^+^ in the form of KCl significantly attenuates this inflammatory response, suggesting that potassium ions modulate the immune reaction triggered by ORF-3a. These results highlight the intricate interplay between potassium ion channels and the inflammatory pathways activated by SARS-CoV-2-ORF-3a. Furthermore, our data show that ORF3a activates caspase-3, a hallmark of apoptosis, which is also reduced by the supplementation of exogenous K^+^. This aligns with the established roles of K^+^ ions in controlling apoptosis and mitochondrial function, emphasizing that the modulation of potassium channels may provide a therapeutic avenue to mitigate both the inflammatory response and apoptosis induced by ORF-3a. The inflammatory response induced by SARS-CoV-2-ORF-3a is diminished when exogenous K^+^ in the form of KCl is added. K^+^ ions are known to regulate cellular signaling pathways, which is consistent with the previous finding that the activation of NLRP3 inflammasomes, a critical modulator of inflammatory reaction, has been linked to K^+^ efflux [77]. In addition, mitochondrial potassium channels such as mitoK^ATP^ and mitoBKCa open, causing mitochondrial depolarization, cytochrome c release, and caspase activation, resulting in apoptosis [78,79,80].

Regarding the upregulation of the ORF-3a protein of *SARS-CoV-2* in the lungs of mice, our results showed that direct expression of this viral protein leads to significant histopathological changes, including inflammatory cell infiltration, alveolar damage, and increased apoptosis (as detected by a TUNEL assay) induced through mitochondrial dysfunction. These results corroborate the previous studies reported by Ren et al. [52]. The IHC for mitoK^ATP^ in the mice lungs transfected with ORF-3 revealed a notable upregulation in MitoK^ATP^ expression. This increased expression of MitoK^ATP^ in response to ORF-3a transfection suggests its involvement in the cellular response. The prominent recruitment of macrophages (f4/80 staining) and neutrophils (Ly6g staining), along with the upregulation of MitoK^ATP^ and IFN-β expression, in the ORF-3a transfection group, suggests that this viral protein may play a critical role in modulating the host’s innate immune response during *SARS-CoV-2* infection. These findings are consistent with previous reports on the immunomodulatory properties of ORF-3a, such as its ability to downregulate MHC-I expression and dysregulate type I interferon signaling [81,82]. The concurrent upregulation of MitoK^ATP^ adds a new dimension to our understanding of ORF-3a’s effects on cellular physiology, highlighting the complex interplay between viral proteins, mitochondrial function, and host immune responses. Overall, this study demonstrates that ORF-3a induces lung damage and apoptosis in mouse models and HeLa cells, but these in vitro and in vivo systems may differ in terms of response mechanisms and cellular environments. The intricate interactions within the whole body, including immunological responses and tissue-specific variables, may not be accurately replicated in HeLa cell cultures, resulting in discrepancies in the apparent effects of ORF-3a.

## 4. Materials and Methods

### 4.1. In Vivo Infection with SARS-CoV-2

The C57BL/6 transgenic (B6/JGpt-H11^emICmK18-ACE2^/Gpt) mice were acquired from gemparhmatech. Co., Ltd. China. All the experiments involving live viruses were conducted with the approval of the Animal Care and Use Committee of Wuhan University (IACUC: protocol number WP20210527) and conducted at Wuhan University in the animal biosafety level 3 (ABSL-3) laboratory. Briefly, mice, 6 to 8 weeks old, were randomly assigned to infection (100 pfu and 1000 pfu) intranasally with the *SARS-CoV-2 wild-type*, *Delta*, or *Omicron variants*. On day 7, mice were euthanized and lung tissues were collected for further analysis.

The mice were transfected with SARS-CoV-2-ORF-3a using an Entranster in vivo DNA transfection reagent from Engreen.China (https://www.engreen.com.cn/) as per the manufacturer’s procedure.

### 4.2. H&E Staining

Lung tissue sections from infected mice were dissected on day 7 after *SARS-CoV-2* infection, fixed in 4% paraformaldehyde, and stained with hematoxylin and eosin (H&E). Briefly, the tissue sections were deparaffinized in xylene and rehydrated through a graded series of ethanol solutions. The sections were stained with Mayer’s hematoxylin for 5 min, followed by a tap water rinse. Next, the sections were differentiated in 1% acid alcohol, rinsed in tap water, and blued in 0.2% ammonia water. After rinsing, the sections were counterstained with eosin for 2 min. Finally, the stained sections were dehydrated through graded alcohols, cleared in xylene, and mounted with a coverslip using a permanent mounting medium [83]. The slides were blindly scanned and analyzed by the Wuhan Pinuofei Technology Company (Wuhan, China) (https://www.pinuofei.com).

### 4.3. TUNEL Assy

Lung tissue was dissected on day 7 after *SARS-CoV-2* infection. Apoptosis in SARS-CoV-2-infected lung tissue was detected by a TUNEL Assay Kit, following the producer’s protocols from the Wuhan Pinuofei Technology Company.

### 4.4. Immunohistochemistry (IHC)

Formalin-fixed, paraffin-embedded lung tissue samples were sectioned and sent to the Wuhan Pinuofei Technology Company for IHC staining. The tissue sections were deparaffinized, rehydrated, and subjected to antigen retrieval. Endogenous peroxidase activity was blocked, and the sections were incubated with primary antibodies targeting the macrophage marker F4/80 and the neutrophil marker Ly6G. After washing, the sections were incubated with appropriate biotinylated secondary antibodies and a streptavidin–horseradish peroxidase complex. The slices were counterstained with hematoxylin, and 3,3′-diaminobenzidine (DAB) was used as the chromogen to achieve visualization [84].

### 4.5. Cell Culture

HeLa cells (Henrietta Lacks cell lines) were acquired from an American-type culture collection (ATCC). They were kept and cultured in Dulbecco’s Modified Eagle’s Medium (DMEM: Gibco) with 10% fetal bovine serum (FBS: Gibco) (Waltham, MA, USA), 100 U/mL penicillin, and 100 μ/mL streptomycin sulfate at 37 °C in a humidified incubator with 5% CO_2_.

### 4.6. Antibodies and Reagents

Antibodies used in the study included Flag (F2464) at 1:2000 dilution and monoclonal anti-GAPDH (G9295) at 1:5000 dilution, both sourced from Sigma (St. Louis, MO, USA). The CCDC51 polyclonal rabbit antibody (20465-1-AP) at 1:2000 dilution and polyclonal rabbit Caspase-3 antibody (19677-1-AP) at 1:1000 dilution were obtained from Proteintech. Additionally, monoclonal antibodies for cytochrome c (250109) were procured from Zanbio, and monoclonal antibodies for Tom-20 (A19403) at a 1:2000 dilution were purchased from ABclonal Technology.

### 4.7. Plasmids and Transfection

The SARS-CoV-2-ORF-3a genes’ full-length DNA fragments were produced by Genscript (Nanjing, China). Subcloning and insertion into mammalian expression vectors of the genes were carried out in the corresponding orders. Lipofectamine 2000 (Invitrogen) was used for plasmid transfections following the producer’s protocols.

### 4.8. Infection with Virus and Collection of Samples

After viral infection, samples were collected and inactivated, including the supernatant, which was treated at 56 °C for 45 min. RNA samples were processed with Trizol reagent, while Western blotting samples were treated with lysis buffer and SDS loading buffer, both heated to 100 °C for 15 min. The inactivated samples were then sent to a biosafety level 2 (BSL-2) facility located at the Hubei Center for Disease Control in Wuhan, China, for subsequent analysis. Additionally, we obtained green fluorescent protein (GFP)-expressing vesicular stomatitis virus (VSV) strains as a kind donation from Dr. Bo Zhong, a researcher affiliated with Wuhan University.

### 4.9. Western Blotting

To extract and purify the proteins, the cells were disrupted in a lysis solution that contained 50 mM Tris-HCl (pH 7.5), 0.5 mM EDTA, 150 mM NaCl, 1% NP40, and 1% SDS. Additionally, a dilution of 1:100 of a phosphatase and protease inhibitor cocktail obtained from Roche was added. The lysed cell mixtures were then stirred at 4 °C for an hour. After this, the supernatant parts were separated, heated in a protein loading buffer for 5 min, and then put through SDS PAGE. To obtain the mitochondrial and cytosolic fractions, cells were treated with digitonin at an optimized concentration to selectively permeabilize the plasma membrane while keeping the mitochondria intact. The cytosolic fraction was collected from the supernatant, and the mitochondrial fraction was obtained by further centrifugation of the remaining pellet. Once the extraction was performed, the samples were transferred to a Polyvinylidene Fluoride membrane and soaked in a blocking solution, with 5% milk in Tris-buffered saline with Tween 20 (TBST) for one hour. Before applying primary antibodies, the membranes were left to soak at 4 °C in TBST mixed with 5% bovine serum albumin (BSA).

Membranes were rinsed with TBST twice for a total of 10 min after incubation. Following this, they were rinsed again after being treated with secondary antibodies conjugated with HRP for one hour. Finally, an X-ray film or LAS-4000 Imager was used to visualize the membranes [85].

### 4.10. Isolation of Mitochondrial and Cytosolic Fractions

HeLa cells were transfected with ORF-3a and then treated with apoptotic inducers, i.e., CCDP and *VSV*. After treatment and incubation, the cells were washed twice with ice-cold PBS and resuspended in an ice-cold isolation buffer (250 mM sucrose, 10 mM HEPES, 1 mM EDTA, pH 7.4) containing protease and phosphatase inhibitors. Digitonin was added to the suspended cells at a 0.02% (*w*/*v*) concentration to permeabilize the plasma membrane selectively. The suspension was then gently mixed on ice for 5 min to allow the selective release of cytosolic contents. The samples were centrifuged at 800× *g* for 5 min at 4 °C to separate the supernatant (cytosolic fraction). The pellet, containing intact mitochondria, was washed with an isolation buffer and lysed in a mitochondrial lysis buffer (50 mM Tris HCl, 10 mM NaCl, 1% NP-40, pH 7.4) for further analysis.

### 4.11. Quantitative RT-PCR Analysis and Primers

Total RNA extraction was performed from the designated cells using the Trizol reagent (Invitrogen Life Technologies, Carlsbad, CA, USA) following the method recommended by the manufacturer. The digitonin method was used to isolate mtRNA [86]. For the synthesis of complementary DNA, 1 µg of RNA was utilized, employing HiScript II qRT Supermix (available from Vazyme Biotech Co., Nanjing, China). Subsequent quantitative real-time analysis utilized ChamQ SYBR qPCR Master Mix (also from Vazyme Biotech Co., Nanjing, China) and was conducted on a Roche LC480 instrument. The PCR reaction mixture was prepared with the following components: 10 µL of SYBR Green PCR master mix, 1 µL of the synthesized DNA, 1 µL of a 10 µM primer solution, and 8 µL of RNase-free water. Gene expression was ascertained using the comparative 2^−∆∆CT^ technique, with GAPDH acting as an internal control for mRNA quantification. qPCR was performed using the following conditions: 5 min at 42 °C, 10 s at 95 °C, 5 s at 95 °C, 40 cycles, and 30 s at 60 °C. The primers used in this study are listed in Table 1.

### 4.12. Chemical Treatment of Cell Lines

HeLa cells were transfected with ORF-3a plasmids. After overnight incubation, different chemical reagents were added to the medium to induce apoptosis: cisplatin (CCDP): Sigma-Aldrich, (10 μM), for 24 h (St. Louis, MO, USA); carbonyl cyanide m-chlorophenyl hydrazone (CCCP): Sigma-Aldrich, (10 μM), for 6 h (St. Louis, MO, USA); and Doxorubicin (Dox): Thermo Fisher Scientific (10 μM) for 24 h (Waltham, MA, USA). All the chemicals were dissolved in DMSO or sterile water. Cells were treated with the vesicular stomatitis virus (VSV: 20 μL, MOI = 1) for 12 h to induce apoptosis in HeLa cells. HeLa cells were treated with KCL (60 mM) to investigate the effect and role of potassium ions in apoptosis.

### 4.13. Fluorescence Microscopy

Cells were maintained in culture dishes for 24 h. Post-transfection, the cells were rinsed twice with 1× PBS. Subsequently, they were stained with the ROS kit (Beyotime, S0033S) and the JC-1 kit (Beyotime, C2006) separately, following the staining protocols recommended by the manufacturer (Shanghai, China).

### 4.14. Apoptosis Assay (ANNEXIN-V Assay)

The apoptosis assay was carried out using the Annexin V-FITC/PI Detection Kit (4A Biotech, Beijing, China) following the manufacturer’s instructions. Briefly, 2 × 10^5^ cells were resuspended in a binding buffer, then stained with 10 μL Annexin V-FITC and 5 μL PI for 15 min. The samples were analyzed by flow cytometry (Beckman Coulter, CytoFLEX LX, Brea, CA, USA).

### 4.15. Flow Cytometry

ROS and mitochondrial membrane potential were detected using the ROS kit (Beyotime, S0033S) and the JC-1 kit (Beyotime, C2006), respectively. The staining was carried out following the directions given by the manufacturer. Flow cytometry was used to assess the cells (Beckman Coulter, cytoflex LX, USA). We conducted trials in triplicate and counted at least 10,000 cells.

### 4.16. Statistical Analysis

All data are presented as means ± standard deviation (SD) to represent variability within groups. Statistical analyses were conducted using Prism 8.3 (GraphPad Software Inc., La Jolla, CA, USA). To compare two groups, unpaired Student’s *t*-tests were used, assuming a normal distribution of data. For comparisons involving more than two groups, a one-way analysis of variance (ANOVA) followed by Bonferroni’s post hoc test was performed to determine statistically significant differences between groups. Additionally, for non-parametric data or when normality could not be assumed, the Kruskal–Wallis test followed by Dunn’s post hoc test was employed. The threshold for statistical significance was set as follows: * *p* < 0.05, ** *p* < 0.01, *** *p* < 0.001 and **** *p* < 0.0001 indicating increasing levels of statistical significance. Non-significant differences were denoted as ns (*p* > 0.05).

## 5. Conclusions

Our study demonstrates that the SARS-CoV-2-ORF-3a protein triggers apoptosis through mitochondrial dysfunction, particularly highlighting the involvement of the potassium (K^+^) ion channel (MitoK^ATP^) in modulating this mitochondrial damage. Our findings demonstrate that ORF-3a not only triggers significant mitochondrial damage but also enhances apoptosis and inflammation in lung tissue. Importantly, the addition of exogenous K^+^ was found to effectively mitigate both the apoptotic and inflammatory responses induced by SARS-CoV-2-ORF-3a, suggesting a therapeutic avenue for intervention.

These findings suggest that targeting potassium ion channels could serve as a promising therapeutic strategy to counteract the detrimental effects of *SARS-CoV-2* infection, potentially reducing severe complications associated with COVID-19. Furthermore, our results indicate that a nuanced approach is essential when considering therapies aimed at ORF-3a, as these strategies must balance the dual roles of this protein in promoting apoptosis and initiating an antiviral immune response. Continued research in this area may lead to the development of selective inhibitors that minimize cellular damage while preserving innate immune function, ultimately contributing to more effective treatments for COVID-19.

## Figures and Tables

**Figure 1 ijms-26-01575-f001:**
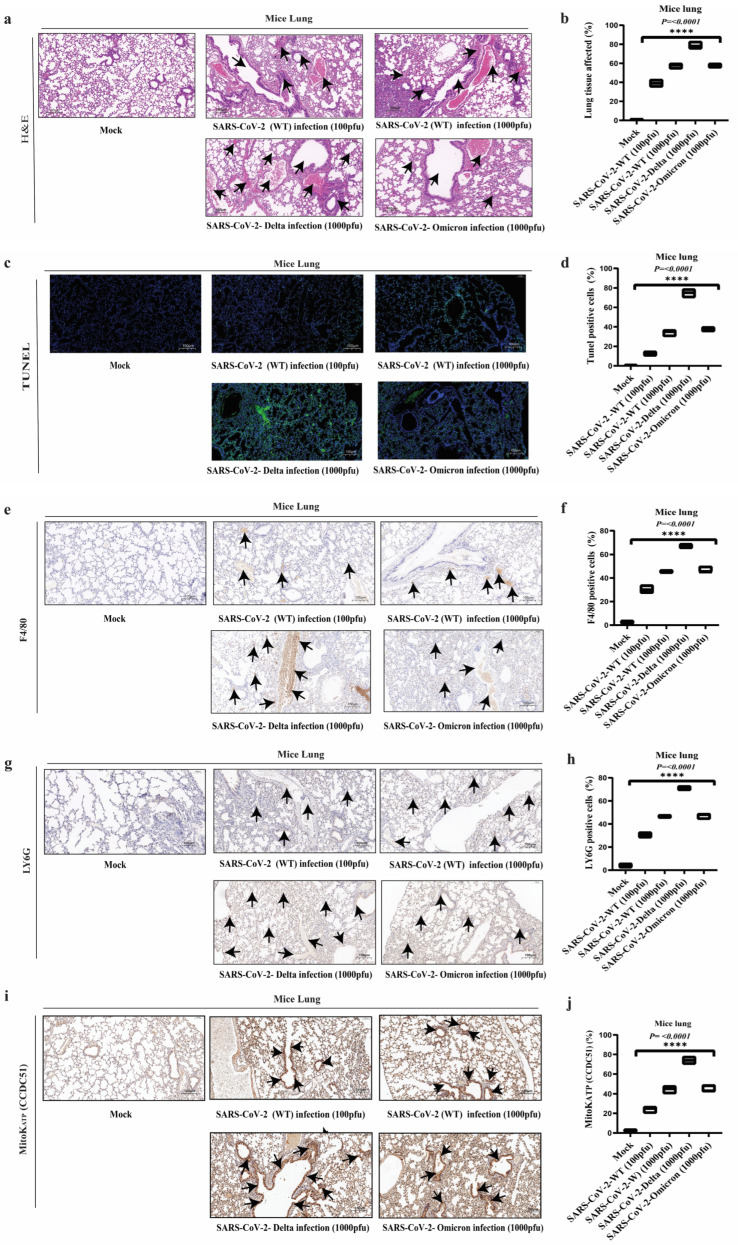
Histopathological and immunohistochemical analysis of mouse lung infections with different *SARS-CoV-2 strains*. (**a**,**b**) Hematoxylin and eosin (H&E) staining of lung tissue sections from mice infected with different *SARS-CoV-2 strains* (*wild type*, *Delta*, and *Omicron BA.2*), showing distinct lung damage patterns. The H&E results revealed significant abnormalities, including cellular morphology changes, nuclear irregularities, eosinophilia, hemorrhage, and increased alveolar space, indicative of pulmonary edema and inflammation. The *Delta variant* caused more severe lung damage compared to *wild-type* and *Omicron variants*, which exhibited milder histopathological changes. Pointed arrows in the histological images highlight these key abnormalities. The histological images are accompanied by representative histograms quantifying lung damage. (**c**,**d**) TUNEL assay of lung tissue sections from mice infected with different *SARS-CoV-2 strains* (*wild type*, *Delta*, and *Omicron BA.2*). The assay highlights apoptotic cell death within the lung tissues, marked by green fluorescence, with significant variation in apoptosis levels observed across different viral strains. Quantitative histograms accompanying the images illustrate the extent of apoptosis, expressed as a percentage of apoptotic nuclei relative to the total nuclei per field. (**e**,**f**) Lung sections from C57BL/6 transgenic mice infected with *wild-type*, *Delta*, and *Omicron variants* were evaluated for F4/80-positive macrophages using immunohistochemistry (IHC). Infected mice showed a significant increase in F4/80-positive macrophages compared to the mock controls. The *Delta variant* displayed the highest macrophage infiltration, followed by *wild-type* and *Omicron variants*. The arrows indicate F4/80-positive macrophages, highlighting their location and distribution within the lung tissue. Immunohistochemical staining of the macrophage marker F4/80 (brown) in lung tissue sections, with corresponding histograms representing macrophage abundance. (**g**,**h**) Lung sections from C57BL/6 transgenic mice infected with *wild-type*, *Delta*, and *Omicron variants* were assessed for Ly6G-positive neutrophils using immunohistochemistry (IHC). Infected mice exhibited a significant increase in Ly6G-positive neutrophils compared to the mock controls. The *Delta variant* showed the highest neutrophil infiltration, indicating a robust inflammatory response, while *wild-type* and *Omicron variants* demonstrated significantly lower levels. Pointed arrows indicate Ly6G-positive neutrophils, emphasizing their location and distribution within the lung tissue. The accompanied histograms quantified neutrophil infiltration in the lungs of infected mice. (**i**,**j**) Immunohistochemistry (IHC) analysis of lung sections from C57BL/6 transgenic mice revealed significantly higher expression of the MitoK^ATP^ channel (brown punctate staining) in *wild-type* and *Delta variant*-infected mice compared to the mock controls, indicating increased mitochondrial stress. The *Omicron variant* displayed lower MitoK^ATP^ expression. The arrows highlight areas of MitoKATP channel expression (brown punctate staining) within the lung tissue, emphasizing its distribution and intensity. Histogram quantifying MitoK^ATP^ channel expression in lung tissue, represented as the average intensity of brown punctate staining per field. Scale bar: 100 μm. Magnification: 10×. Data are representative of *n* = 3 animals per group. Statistical significance was assessed using the Kruskal–Wallis test, comparing the infected groups to the control. Significant differences are indicated as **** *p* < 0.0001.

**Figure 2 ijms-26-01575-f002:**
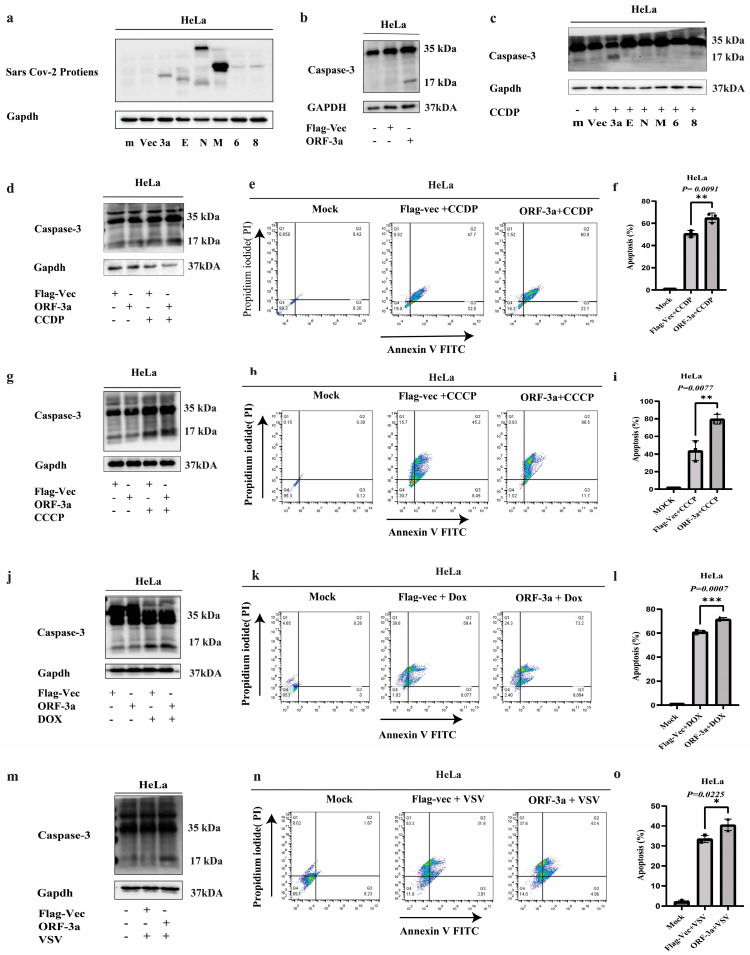
SARS-CoV-2-ORF-3a-induced apoptosis: (**a**) Western blot analysis of HeLa cells transfected with various SARS-CoV-2 viral proteins (ORF-3a, E, M, N, 6, and 8), along with Flag-Vec and mock (m). (**b**) Western blot analysis of cleaved caspase-3 shows that ORF-3a induces apoptosis in HeLa cells compared to Flag-Vec and mock. (**c**) Expression of cleaved caspase-3 following 6 h treatment with CCDP in HeLa cells transfected with SARS-CoV-2 proteins. (**d**) Western blot analysis of cleaved caspase-3 in HeLa cells transfected with ORF-3a and Flag-Vec and treated with CCDP for 6 h, confirming that ORF-3a promotes apoptosis. (**e**,**f**) Flow cytometric analysis of Annexin V-FITC/PI-stained HeLa cells treated with CCDP for 6 h, showing increased apoptotic population in cells transfected with ORF-3a compared to Flag-Vec. The apoptotic cell percentage is shown in the accompanying histogram. (**g**–**i**) Western blot and flow cytometry analyses showed that CCCP treatment for 30 min enhances apoptosis in ORF-3a-transfected HeLa cells, as indicated by increased cleaved caspase-3 expression and Annexin-V-positive cells. Apoptotic cell percentage is shown in histogram. (**j**–**l**) Western blot and flow cytometry analysis of cleaved caspase-3 and apoptotic cell frequency following 24 h Doxorubicin (Dox) treatment, demonstrating that ORF-3a sensitizes HeLa cells to apoptosis. Apoptotic cell percentage is shown in histogram. (**m**–**o**) Western blot and flow cytometric analysis showing that VSV infection (20 μL, MOI = 1, 12 h) increases cleaved caspase-3 expression and apoptotic cell population in HeLa cells transfected with ORF-3a. Apoptotic cell percentage is shown in histogram. Data are presented as mean ± SEM. Statistical significance was determined by unpaired *t*-test: * *p* < 0.05, ** *p* < 0.01, and *** *p* < 0.001.

**Figure 3 ijms-26-01575-f003:**
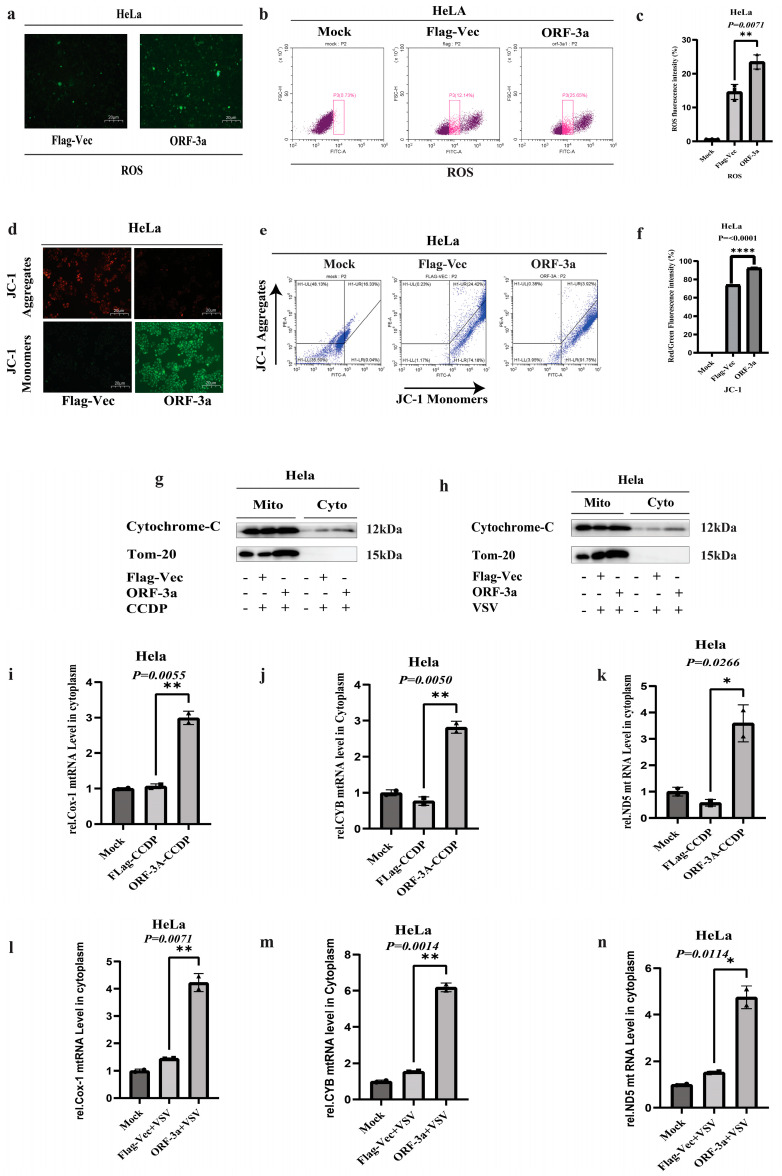
SARS-CoV-2-ORF-3a induces mitochondrial dysfunction: (**a**) Fluorescence microscopy images showing reactive oxygen species (ROS) production in HeLa cells transfected with SARS-CoV-2 ORF-3a or Flag-vector control. Increased green fluorescence in ORF-3a-transfected cells indicates elevated ROS levels compared to the control. (**b**,**c**) Flow cytometry analysis quantifying ROS levels in HeLa cells transfected with SARS-CoV-2 ORF-3a. A significant increase in ROS production is observed in ORF-3a-transfected cells compared to Flag-vector controls. (**d**) Fluorescence microscopy images of JC-1-stained HeLa cells, indicating loss of mitochondrial membrane potential (Δψm) in ORF-3a-transfected cells. This is evidenced by a decrease in red fluorescence and an increase in green fluorescence. (**e**,**f**) Flow cytometry analysis of Δψm in JC-1-stained cells. A decreased red/green fluorescence ratio in ORF-3a-transfected cells suggests a significant loss of Δψm. (**g**,**h**) Western blot analysis showing the release of cytochrome c (cyt-c) into the cytosol in HeLa cells transfected with SARS-CoV-2 ORF-3a and treated with CCDP and VSV. A significant increase in cyt-c levels is observed in the cytosolic fraction of ORF-3a-transfected cells. (**i**–**k**) qPCR analysis of mitochondrial gene expression (Cox-1, CYB, and ND5) in HeLa cells transfected with ORF-3a and treated with CCDP. A significant upregulation of these mitochondrial genes is observed in ORF-3a-transfected cells compared to controls. (**l**–**n**) qPCR analysis of mitochondrial gene expression (Cox-1, CYB, and ND5) in HeLa cells treated with VSV (20 μL, MOI = 1). ORF-3a-transfected cells show significant upregulation of these mitochondrial genes in response to VSV infection. All gene expression data were normalized to GAPDH as an internal control. Data are presented as mean ± SEM. Statistical significance was determined using an unpaired *t*-test. Values of * *p* < 0.05, ** *p* < 0.01, and **** *p* < 0.0001 indicate statistical significance compared to control. Scale bar: 20 μm.

**Figure 4 ijms-26-01575-f004:**
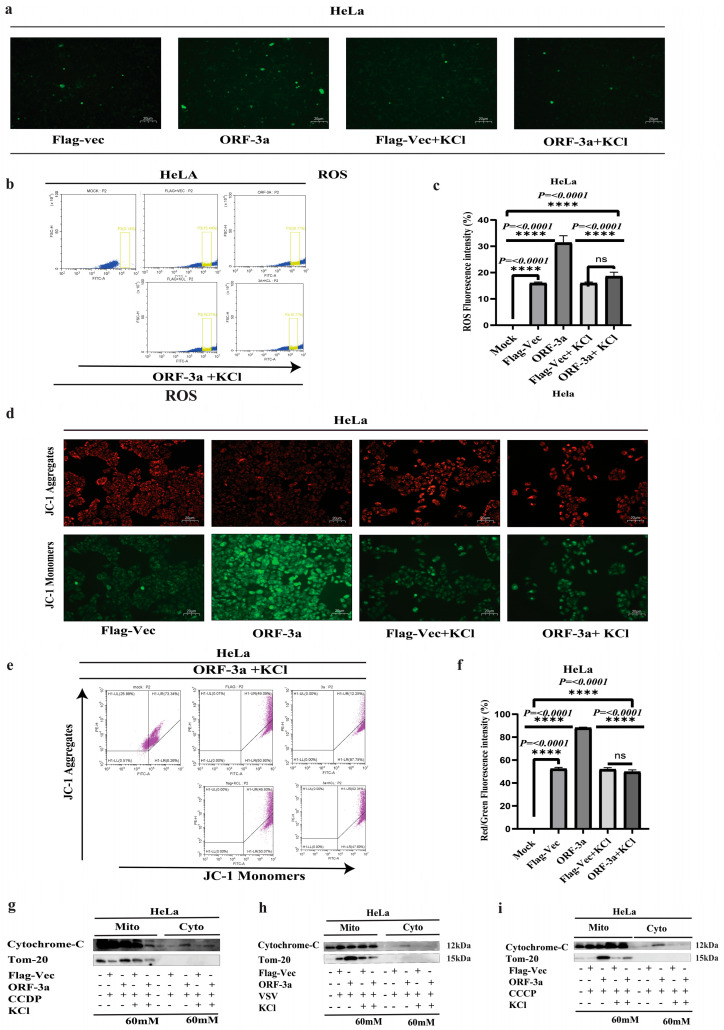
SARS-CoV-2-ORF-3a mediates mitochondrial dysfunction through the K^+^ ion channel: (**a**) Fluorescence microscopy images showing reactive oxygen species (ROS) production in HeLa cells transfected with SARS-CoV-2 ORF-3a and treated with exogenous potassium (KCl, 60 mM). The addition of K^+^ significantly reduced ROS generation compared to ORF-3a-transfected cells without K^+^ supplementation. (**b**,**c**) Flow cytometry analysis of ROS levels in ORF-3a-transfected HeLa cells treated with exogenous K^+^ (KCl, 60 mM). A significant decrease in ROS production was observed in cells treated with K^+^ compared to ORF-3a-transfected cells without K^+^ supplementation. (**d**) Fluorescence microscopy images of JC-1-stained HeLa cells, showing reduced mitochondrial membrane potential (Δψm) and depolarization in cells treated with exogenous K^+^ compared to ORF-3a-transfected cells alone. This suggests that K^+^ supplementation mitigates mitochondrial damage. (**e**,**f**) Flow cytometric analysis of JC-1 fluorescence in HeLa cells transfected with ORF-3a and treated with exogenous K^+^. A significant decrease in green fluorescence (indicative of JC-1 monomers) and an increase in red fluorescence (indicative of JC-1 aggregates) were observed, indicating reduced mitochondrial depolarization in the presence of K^+^. (**g**–**i**) Western blot analysis showing a decrease in cytochrome c (cyt-C) release into the cytosol in ORF-3a-transfected HeLa cells treated with CCDP, VSV (MOI = 1), and CCCP after supplementation with exogenous K^+^. These results suggest that K^+^ supplementation mitigates mitochondrial dysfunction and apoptosis signaling. Data are presented as mean ± SEM. Statistical significance was assessed using an ANOVA followed by a Bonferroni post hoc test. **** *p* < 0.0001 indicate statistical significance.

**Figure 5 ijms-26-01575-f005:**
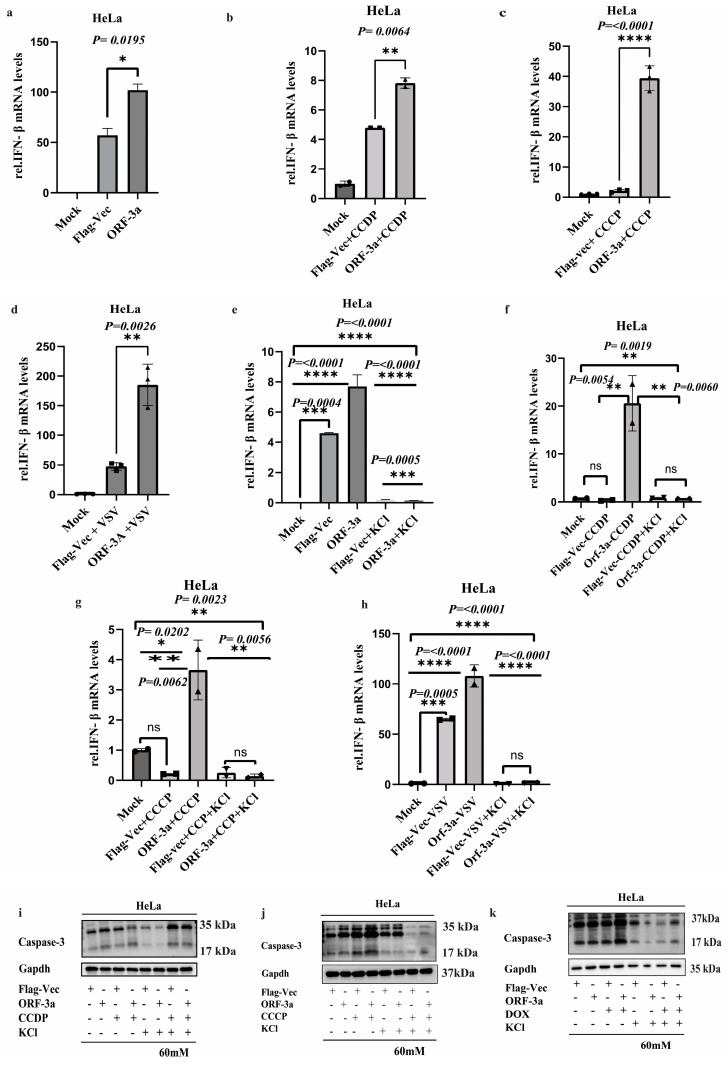
Exogenous K^+^ attenuates SARS-CoV-2-ORF-3a-induced inflammatory responses and apoptosis: (**a**) qPCR analysis of IFN-β mRNA expression in HeLa cells transfected with SARS-CoV-2-ORF-3a, Flag-Vec, and mock alone. ORF-3a significantly increased IFN-β expression compared to the Flag-Vec and mock groups, indicating activation of the innate immune response. (**b**–**d**) qPCR analysis of IFN-β expression in HeLa cells transfected with ORF-3a and treated with apoptotic inducers (CCDP, CCCP, and VSV, MOI = 1). Treatment with apoptotic inducers resulted in a significant increase in IFN-β expression in ORF-3a-transfected cells compared to Flag-Vec and the mock control. (**e**) Effect of exogenous K^+^ supplementation on IFN-β expression in HeLa cells transfected with ORF-3a, Flag-Vec, and mock. Supplementation with KCl resulted in a significant reduction in IFN-β mRNA expression in ORF-3a-transfected cells, suggesting that potassium ions modulate the inflammatory response induced by SARS-CoV-2-ORF-3a. (**f**–**h**) qPCR analysis of IFN-β expression in HeLa cells transfected with ORF-3a and exposed to apoptotic inducers (CCDP, CCCP, and VSV, MOI = 1), followed by treatment with KCl. The addition of exogenous K^+^ significantly suppressed IFN-β expression compared to Flag-Vec and mock cells, highlighting the role of potassium ion channels in modulating the inflammatory response during apoptotic stress. (**i**–**k**) Western blot analysis of cleaved caspase-3 expression in HeLa cells transfected with ORF-3a and exposed to apoptotic inducers (CCDP, CCCP, and VSV), followed by treatment with KCl. A significant reduction in cleaved caspase-3 expression was observed in KCl-treated ORF-3a cells compared to cells treated with apoptotic inducers alone, suggesting that potassium supplementation mitigates apoptosis induced by SARS-CoV-2-ORF-3a. Data were analyzed using appropriate statistical tests, including unpaired *t*-tests and ANOVAs followed by Bonferroni post hoc tests. Statistical significance was defined as * *p* < 0.05, ** *p* < 0.01, *** *p* < 0.001 and **** *p* < 0.0001.

**Figure 6 ijms-26-01575-f006:**
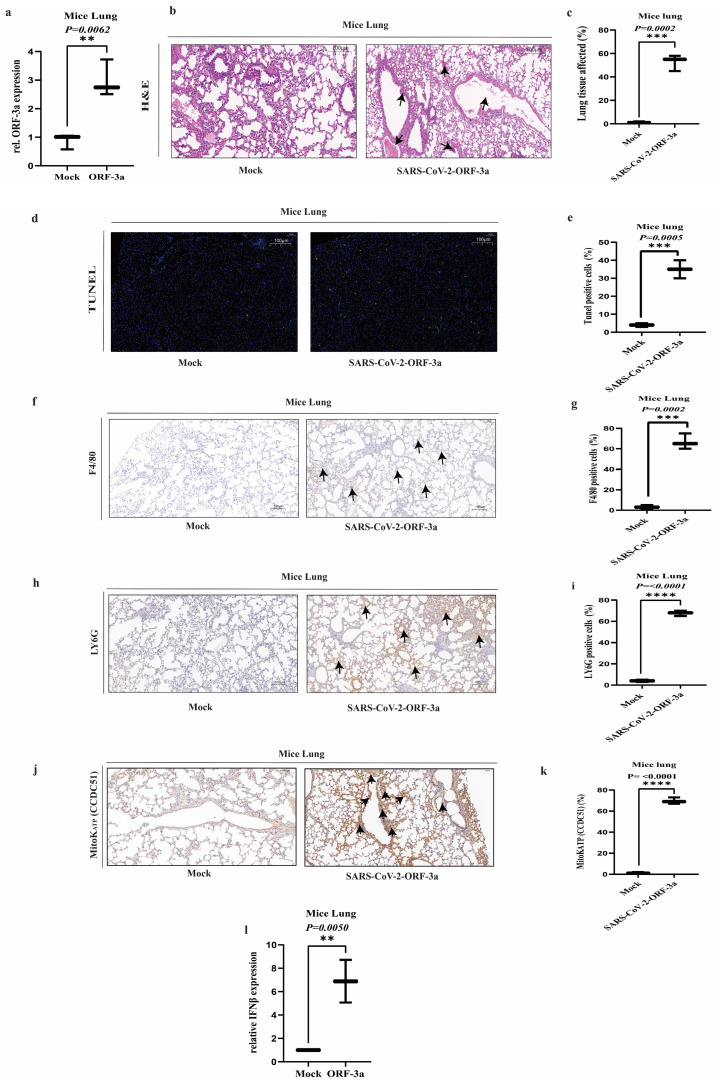
Lung histopathology and immune cell infiltration in mice transfected with SARS-CoV-2-ORF-3a protein: (**a**) qPCR analysis of ORF-3a mRNA expression in lung tissue of C57BL/6 mice transfected with SARS-CoV-2 ORF-3a and mock. ORF-3a expression was significantly higher in the lungs of transfected mice compared to mock mice, confirming successful transfection and expression of ORF-3a in lung tissue. (**b**,**c**) Representative H&E images of lung sections from SARS-CoV-2 ORF-3a-transfected and mock mice. The ORF-3a group displayed widespread tissue damage, characterized by inflammatory infiltrates and alveolar damage, whereas the mock group showed normal lung architecture. The pointed arrows highlight key histopathological features, including inflammatory cell infiltration, disrupted alveolar structures, and areas of tissue damage observed in the ORF-3a group. (**d**,**e**) TUNEL assay of lung sections from mice transfected with SARS-CoV-2 ORF-3a and mock. The TUNEL assay revealed significantly increased apoptotic cells in the ORF-3a-transfected group, indicating that ORF-3a induces apoptosis in lung tissue. (**f**,**g**) Immunohistochemistry (IHC) analysis of F4/80-positive macrophages in lung tissue from mice transfected with SARS-CoV-2 ORF-3a and mock. A significant increase in F4/80-positive macrophages was observed in the ORF-3a-transfected group, indicating macrophage recruitment in response to inflammation. The arrows highlight F4/80-positive macrophages (brown staining), emphasizing their location and distribution within the lung tissue. (**h**,**i**) IHC analysis of Ly6G-positive neutrophils in lung tissue from SARS-CoV-2 ORF-3a-transfected and mock mice. A significant increase in Ly6G-positive neutrophils was observed in the ORF-3a-transfected mice, further supporting the activation of an inflammatory response. The arrows indicate the Ly6G-positive neutrophil (brown staining), representing their locality and distribution within the lung tissue transfected with ORF-3a. (**j**,**k**) IHC analysis of MitoK^ATP^ (CCDC51) expression in lung tissue from SARS-CoV-2 ORF-3a-transfected and mock mice. Enhanced punctate staining of MitoK^ATP^ was observed in ORF-3a-transfected mice, indicating increased mitochondrial stress in response to ORF-3a expression. The pointed arrows highlight areas of MitoK^ATP^ expression, emphasizing the distinct punctate staining patterns observed in the lung tissue transfected with ORF-3a. (**l**) qPCR analysis of interferon-beta (IFN-β) expression in lung tissue of C57BL/6 mice transfected with SARS-CoV-2 ORF-3a and mock. The mRNA expression of IFN-β was significantly higher in the lungs of ORF-3a-transfected mice compared to mock mice, suggesting activation of the innate immune response. Data were analyzed using appropriate statistical tests, including unpaired *t*-tests. Statistical significance was defined as ** *p* < 0.01, *** *p* < 0.001 and **** *p* < 0.0001.

**Table 1 ijms-26-01575-t001:** Primers used in this study.

Mouse GAPDH Forward: 5′-AGGTCGGTGTGAACGGATTTG-3′	Mouse GAPDH Reverse: 5′-GGGGTCGTTGATGGCAACA-3′
Human GAPDH Forward: 5′-GGAGCGAGATCCCTCCAAAAT-3′	Human GAPDH Reverse: 5′-GGCTGTTGTCATACTTCTCATGG-3′
Mouse IFN-β Forward: 5′-AGATCAACCTCACCTACAGG-3′	Mouse IFN-β Reverse: 5′-TCAGAAACACTGTCTGCTGG-3′
Human IFN-β Forward: 5′-ATGACCAACAAGTGTCTCCTCC-3′	Human IFN-β Reverse: 5′-GGAATCCAAGCAAGTTGTAGCTC-3′
SARS-CoV-2-ORF-3a Forward: 5′-GAGATGGCAACTAGCACTCT-3′	SARS-CoV-2-ORF-3a Reverse: 5′-AGAAAAGGGGCTTCAAGGCC-3′
Human-COX-1 Forward: 5′-ACGTTGTAGCCCACTTCCAC-3′	Human-COX1 Reverse: 5′-TGGCGTAGGTTTGGTCTAGG-3′
Human-ND5 Forward: 5′-TCGAAACCGCAAACATATCA-3′	Human-ND5 Reverse: 5′-CAGGCGTTTAATGGGGTTTA-3′
Human-CYB Forward: 5′-AGACAGTCCCACCCTCACAC-3′	Human-CYB Reverse: 5′-GGTGATTCCTAGGGGGTTGT-3′

## Data Availability

The datasets used and analyzed during the current study are available from the corresponding author upon reasonable request.

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
