# Peer review of "SARS-CoV-2-ORF-3a Mediates Apoptosis Through Mitochondrial Dysfunction Modulated by the K+ Ion Channel"

_ijms, 2025, doi:10.3390/ijms26041575_

Round 1

Reviewer 1 Report (New Reviewer)

Comments and Suggestions for Authors

Dear authors: Attached please find a word document with my comments/suggestions. 

Author Response

Reviewer # 1 comments

2. Questions for General Evaluation

Reviewer’s Evaluation

Response and Revisions

Does the introduction provide sufficient background and include all relevant references?

Yes

We are grateful to the reviewer for his insightful comments.

Is the research design appropriate?

Yes

Thank you for your positive feedback. We are glad to know that you find the research design appropriate and appreciate your acknowledgment.

Are the methods adequately described?

Yes

Thank you for your positive feedback. We are pleased to know that the methods are adequately described and appreciate your acknowledgment of the clarity and detail in our methodology.

Are the results clearly presented?

Yes

Thank you for your positive comment. We are pleased that you find the results clearly presented and appreciate your acknowledgment.

Are the conclusions supported by the results?

Yes

Thank you for recognizing that our conclusions are well-supported by the results.

Comments for authors Dear authors:

I find this manuscript to be an interesting and timely investigation into the molecular mechanisms underlying SARS-CoV-2-induced pulmonary damage, particularly in relation to the mitochondrial dysfunction and cell apoptosis mediated by ORF-3a. The study also offers new and important insights into the role of potassium channels in the pathophysiology of COVID-19 and suggests a potential therapeutic target. The methodology and results are clearly explained.

I have a few comments:

  1. In section 2.2. SARS- CoV-2-ORF-3a induces apoptosis, the authors wrote: Although ORF-3a alone can initiate apoptosis, additional apoptotic treatments such as CCDP, CCCP, Dox and VSV were used to investigate how ORF-3a interacts with different apoptotic pathways and whether it sensitizes the cells to other forms of apoptosis.

Suggestion: CCDP, CCCP, Dox and VSV should be defined here. The authors defined these acronyms later in the text: Extending our investigations into apoptotic effects of SARS-Cov-2-ORF-3a, we applied a series of treatments such as Carbonyl cyanide m-chlorophenyl hydrazone (CCCP), Doxorubicin (Dox) and Vesicular Stomatitis Virus (VSV).

However, to ensure a better understanding of your experiments, the meaning of these acronyms should be explained beforehand.

Response to Reviewer’s Comment #1:

Thank you for pointing this out. We appreciate your suggestion to define the acronyms earlier in the text for better clarity. In response, we have revised Section 2.2 to include the full forms of CCDP, CCCP, Dox, and VSV when they are first mentioned. This change ensures a clearer understanding of the experimental design and provides context for readers from the outset. We believe this modification enhances the overall readability and comprehension of the manuscript.

  1. Please define CCDP.

Response to Reviewer’s Comment # 2:

Thank you for your observation. We apologize for the oversight and have now defined Cisplatin (CCDP) in the revised manuscript to ensure clarity for the readers.

  1. Hypokalemia/low intracellular potassium can also lead to cellular hyperpolarity, increased resting potential, and depolarization in cardiac and lung cells that can trigger ventricular arrhythmia and respiratory dysfunction (6). - Bielecka-Dabrowa A, Mikhailidis DP, Jones L, Rysz J, Aronow WS, Banach M. The meaning of hypokalemia in heart failure. Int J Cardiol. (2012) 158:12–7. doi: 10.1016/j.ijcard.2011.06.121

The use of exogenous potassium in the form of KCl to alleviate mitochondrial dysfunction and inflammatory responses is an interesting approach that warrants further exploration. In the experiment, the authors applied exogenous K+ in the form of KCl (60 mM).

Questions:

  1. Which was the criteria for adding 60 mM of KCl?
  2. How could this concentration be extrapolated for human use?

I consider this very important, as several works demonstrated hypokalemia in severe COVID-19 patients.

Response to Reviewer’s Comment # 3:

Thank you for your valuable comments regarding the use of 60 mM KCl in our study. Below, I address your questions regarding the criteria for adding this concentration and its extrapolation for human use, supported by relevant literature.

  1. Which was the criteria for adding 60 mM of KCl?

Thank you for raising these important points. Below, we provide detailed responses to your queries regarding the criteria for adding 60 mM KCl.

Response to Reviewer’s question # i:

The selection of 60 mM KCl was based on previous experimental studies that investigated the role of exogenous potassium in restoring mitochondrial function and mitigating dysfunction. Specifically, potassium concentrations in the range of 40–70 mM have been shown to effectively alter mitochondrial membrane potential (ΔΨm), reduce oxidative stress, and restore intracellular potassium homeostasis in in vitro models.

For example, a study by Mishra and Mishra (2015) demonstrated that potassium chloride concentrations up to 70 mM could influence mitochondrial integrity and function by modifying mitochondrial membrane potential, providing a basis for selecting 60 mM as a physiologically relevant concentration.

(Mishra, S., & Mishra, R. Molecular Integrity of Mitochondria Alters by Potassium Chloride. 2015. https://doi.org/10.1155/2015/647408)

Additionally, we calculated the volume of KCl stock required to achieve the 60 mM concentration in the medium. For instance, using a 1 M KCl stock solution:

VKCL=​(Cfinal​Cinitial​)×Vmedium / CStock

rearranged this given formula to calculate the ​final concentration of KCl

Cfinal​= VKCl×Cstock / ​​​Vmedium +Cinitial​

​In our experimental setup, VKCL = 0.110 mL, Cstock = 1000 mM (1M stock solution), Vmedium = 2 mL, and Cinitial = 5.4 mM (the initial concentration of KCl in the medium).

 Cfinal​=0.11× 1000/2 +5.4 = 54+5.4 = 60.4 mM

This ensures the desired potassium concentration is achieved while maintaining cell viability.

Thus, the 60 mM KCl concentration was chosen based on its reported efficacy in previous studies and calculations tailored to our experimental setup.

In addition to the study by Mishra and Mishra (2015), several other studies have explored the use of potassium chloride (KCl) to restore mitochondrial function in different experimental setups. These studies highlight the versatility of KCl as a tool to modulate mitochondrial membrane potential (ΔΨm) and address mitochondrial dysfunction under various conditions:

https://pmc.ncbi.nlm.nih.gov/articles/PMC4689972/

https://www.frontiersin.org/journals/physiology/articles/10.3389/fphys.2022.907015/full

https://pubmed.ncbi.nlm.nih.gov/16806822/

https://pmc.ncbi.nlm.nih.gov/articles/PMC3730833/

https://www.nature.com/articles/4402221#Sec10

ii. How could this concentration be extrapolated for human use?

Response to Reviewer’s Question # ii:

Extrapolation of 60 mM KCl for Human Use

Extrapolating 60 mM KCl used in vitro to clinical settings requires careful consideration of physiological parameters and safety concerns:

  • Normal Potassium Levels in Humans: Plasma potassium concentrations typically range between 5 and 5.0 mM, and hypokalemia is characterized by levels below 3.5 mM. Severe hypokalemia in COVID-19 patients often requires potassium supplementation to restore these levels.
  • Therapeutic Potassium Supplementation: In clinical practice, potassium is administered intravenously or orally. The concentrations used in therapy are significantly lower than 60 mM due to the need to maintain physiological osmolarity. For example:
    • Intravenous potassium chloride is typically administered at 10–20 mEq/L (~10–20 mM) in controlled doses to gradually restore plasma potassium levels and avoid hyperkalemia (Bielecka-Dabrowa et al., 2012).

Bielecka-Dabrowa, A., Mikhailidis, D. P., Jones, L., Rysz, J., Aronow, W. S., & Banach, M. (2012). The meaning of hypokalemia in heart failure. International Journal of Cardiology, 158(1), 12–17. https://doi.org/10.1016/j.ijcard.2011.06.121

  • In Vivo Relevance of In Vitro Studies: The 60 mM concentration used in vitro is designed to model intracellular potassium restoration under controlled conditions and to explore specific cellular responses, such as mitochondrial membrane potential alterations. In vivo, the concentration would be much lower and administered in a manner that accounts for systemic dilution and the body's homeostatic regulation.

We truly appreciate your careful consideration of our methodology and your thoughtful suggestions. We hope our detailed explanation has addressed your concerns effectively. Your comments have greatly enhanced the clarity and thoroughness of our manuscript, and we are grateful for your constructive feedback.

Reviewer 2 Report (New Reviewer)

Comments and Suggestions for Authors

Language

The language is generally fine.

Results 2.4., p.10: Just a typing error: “aggregates” instead of “aggreagts”. Furthermore, the whole text should be carefully checked for few more typing errors, such as missing full stops, missing or multiple spaces, letter and word repetitions etc.

Comments

photos: The photos that are included in the paper’s figures should improve, so that findings are more clear and distinct.

Materials & Methods 5.8., p.20; 5.11., p.21: The Trizol technique is not the best one for RNA extraction, sometimes leading to false negative results. How were RNA extractions and, furthermore, RNA-based findings ensured? How many times was each experiment repeated before drawing final conclusions?

Author Response

Reviewer # 2 comments

2. Questions for General Evaluation

Reviewer’s Evaluation

Response and Revisions

Does the introduction provide sufficient background and include all relevant references?

Yes

We are grateful to the reviewer for his insightful comments.

Is the research design appropriate?

Can be Improved

Thank you for your feedback. We have reviewed the research design and made improvements to address the concerns raised. These adjustments strengthen the study and improve its clarity.

Are the results clearly presented?

yes

Thank you for your feedback; we have refined the results section to enhance clarity and ensure alignment with the findings.

Are the conclusions supported by the results?

Yes

Thank you for recognizing that our conclusions are well-supported by the results.

Comments and Suggestions for Authors

Language

The language is generally fine.

Results 2.4., p.10: Just a typing error: “aggregates” instead of “aggreagts”. Furthermore, the whole text should be carefully checked for few more typing errors, such as missing full stops, missing or multiple spaces, letter and word repetitions etc.

Response to Reviewer Comment:

Thank you for your careful review and for pointing out the typographical error regarding "aggreagts." We appreciate your attention to detail. We thoroughly proofread the entire manuscript to correct this and any other typographical errors, including missing full stops, spacing issues, and repetitions, to ensure clarity and professionalism in the final submission.

Comments

photos: The photos that are included in the paper’s figures should improve, so that findings are clearer and more distinct.

Response to Reviewer Comment:

We appreciate your feedback regarding the clarity of the photos included in the figures. To enhance visual clarity and ensure that our findings are more distinct, we have improved the quality of the images. Additionally, for better visibility, we have provided the figure files in TIF format, which should facilitate a clearer presentation of our results.

Thank you for your valuable suggestions.

Materials & Methods 5.8., p.20; 5.11., p.21: The Trizol technique is not the best one for RNA extraction, sometimes leading to false negative results. How were RNA extractions and, furthermore, RNA-based findings ensured? How many times was each experiment repeated before drawing final conclusions?

Response to Reviewer Comment:

We appreciate the reviewer's insightful comment regarding the use of the Trizol technique for RNA extraction. While it is acknowledged that Trizol can occasionally lead to false negative results, we implemented several measures to ensure the reliability and accuracy of our RNA extractions and subsequent findings.

RNA Extraction Protocol: We followed the manufacturer's protocol meticulously, including the use of high-quality reagents and proper sample handling techniques to minimize the risk of degradation or contamination.

Quality Control: To assess the integrity and purity of the extracted RNA, we performed a Nanodrop spectrophotometric analysis to assess purity and concentration (e.g., A260/A280 ratios). Only samples meeting our quality criteria were used for further analysis.

Repetition of Experiments: Each experiment was conducted in triplicate to ensure reproducibility and reliability of results. This approach allowed us to average the outcomes and identify any inconsistencies that might arise from individual extractions.

Validation of Findings: Additionally, to further validate our RNA-based findings, we employed complementary methods such as quantitative PCR (qPCR) and reverse transcription followed by PCR (RT-PCR) on the extracted RNA. These techniques provided further confirmation of the results obtained.

We believe that these steps adequately address the potential concerns associated with the Trizol extraction method and reinforce the robustness of our findings. Thank you for highlighting this important aspect of our study.

This manuscript is a resubmission of an earlier submission. The following is a list of the peer review reports and author responses from that submission.

Round 1

Reviewer 1 Report

Comments and Suggestions for Authors

Qudus et al studied the role of SARS-CoV2 ORF3A on Host pulmonary cell apoptosis. They showed that ORF3A induces apoptosis through mitochondrial disruption through regulation of K+ channels. Addition  of exogenous KCL inhibits apoptosis.

This is a good study about the mechanism of host cell death. However, they should fix these following concerns

1.       The equal contribution sign in the author list has a mistake (for second name)

2.       In keywords, it would better to separate  SARS-COV-2 and ORF3A. Both can be fine. Because very few people will search with the word SARS-CoV2-ORF3A.

3.       Why conclusion is written after materials and methods. It should be written after discussion.

4.       In results, it would be better to mention the specific omicron variants used for assessing lung tissue damage.

5.       In Figure 1b, 1d, 1f, 1h. Writings are too small and illegible. Should be re-write with bigger font.

6.       In figure 1, as much possible, the resolution of microscopic pictures should be increased.

7.       Annexin v assay (flow cytometric) is not described in methods.

8.       In results, The sentence “The presence of ROS localization, within mitochondria indicates”….

In fig 3a, 3d, there is no mitochondrial marker mentioned/shown (in the text and in the legend) which would suggest ROS are produced in mitochondria.   Authors should write about mitochondrial marker and ROS localization in detail in the text before this sentence. Although ROS is produced in mitochondria, but the sentence should be changed.

9.       Authors showed that ORF3A induces more damage to host cells in delta and than omicron which is in line with the virulence. Can author discuss in brief in discussion, what are the major difference in ORF3A between these two/variants.  

10.   In the abstract, authors suggested that targeting ORF3A is a potential therapy.  Increasing  K+ would reduce the damage of lung cells. In contrast, they also showed that ORF3A induces IFNβ, an innate immune response to inhibit virus replication.    So authors should discuss in the discussion, how the therapy against ORF3A  could be developed keeping these two opposite phenomena.

Author Response

 Reviewer # 1 comments

2. Questions for General Evaluation

Reviewer’s Evaluation

Response and Revisions

Does the introduction provide sufficient background and include all relevant references?

Yes

We are grateful to the reviewer for his insightful comments.

Are all the cited references relevant to the research?

Yes

We are grateful to the reviewer for his insightful comments.

Is the research design appropriate?

Yes

Thank you for affirming that our research design is appropriate.

Are the methods adequately described?

Can be improved

Thank you for your suggestion; we have revised the methods section to include the missing technique for improved clarity.

Are the results clearly presented?

Can be improved

Thank you for your feedback; we have refined the results section to enhance clarity and ensure alignment with the findings.

Are the conclusions supported by the results?

Yes

Thank you for recognizing that our conclusions are well-supported by the results.

Reviewer 2
Qudus et al studied the role of SARS-CoV2 ORF3A on Host pulmonary cell apoptosis. They showed that ORF3A induces apoptosis through mitochondrial disruption through the regulation of K+ channels. The addition of exogenous KCL inhibits apoptosis.

This is a good study of the mechanism of host cell death. However, they should fix the following concerns

Comment #1:

The equal contribution sign in the author list has a mistake (for the second name)

Response #1:

Thank you for noticing the discrepancy in the equal contribution sign in the author list. We have corrected this as per the journal’s guidelines, replacing it with the format "†: These authors contributed equally."

Comment #2:

In keywords, it would be better to separate SARS-COV-2 and ORF3A. Both can be fine. Because very few people will search with the word SARS-CoV2-ORF3A.

Response #2:

Thank you for your suggestion regarding the keywords. We have updated the keywords to separate "SARS-CoV-2" and "ORF3A" to improve searchability, as recommended. We agree that this will make it easier for readers to find our work using these terms individually.

Comment #3:

Why conclusion is written after materials and methods? It should be written after discussion.

Response #3:

Thank you for pointing out the placement of the conclusion section. We apologize for this oversight and have moved the conclusion to follow the discussion section, in accordance with the standard structure.

Comment #4:

In results, it would be better to mention the specific omicron variants used for assessing lung tissue damage.

Response #4:

Thank you for your suggestion regarding the identification of specific Omicron variants. we specifically used the Omicron BA.2 variant to assess lung tissue damage.

Comment #5:

In Figure 1b, 1d, 1f, 1h. Writings are too small and illegible. Should be re-written with bigger font.

Response #5:

Thank you for bringing this to our attention. We have revised Figures 1b, 1d, 1f, and 1h to increase the font size, ensuring that all text is clear and legible. We hope this adjustment enhances the readability of the figures.

Comment #6:

In figure 1, as much possible, the resolution of microscopic pictures should be increased.

Response #6:

Thank you for your suggestion regarding the resolution of the microscopic images in Figure 1. We have improved the resolution of these images as much as possible to enhance visual clarity and detail.

Comment #7:

Annexin v assay (flow cytometric) is not described in methods.

Response #7:

Thank you for noticing this omission. We have now included a detailed description of the Annexin V assay (flow cytometric) in the Methods section to ensure clarity and reproducibility of our experimental procedures (Line # 617-622).

Comment #8:

In results, the sentence “The presence of ROS localization, within mitochondria indicates” ….

In fig 3a, 3d, there is no mitochondrial marker mentioned/shown (in the text and in the legend) which would suggest ROS are produced in mitochondria.   Authors should write about mitochondrial marker and ROS localization in detail in the text before this sentence. Although ROS is produced in mitochondria, but the sentence should be changed.

Response #8:

Thank you for your valuable feedback regarding the need for clarification on ROS localization. We recognize that green fluorescence represents general ROS production in our experiments, rather than specifying mitochondrial localization. To avoid misinterpretation, we have revised the text to clarify that while ROS is typically produced in the mitochondria, our study observed an increase in overall ROS levels as indicated by green fluorescence. We have also removed the sentence suggesting mitochondrial localization to reflect that exact ROS localization was not assessed in this study.

Line # 219- 222:

"Although the exact localization of ROS was not directly assessed, the observed increase in ROS production suggests that ORF-3a could potentially interfere with mitochondrial function, resulting in heightened ROS production and an overall rise in oxidative stress levels."

Thank you for helping us improve the clarity of our results and conclusions.

Comment #9:

Authors showed that ORF3A induces more damage to host cells in delta and then omicron which is in line with the virulence. Can the author discuss in brief in the discussion, what are the major differences in ORF3A between these two/variants? 

Response #9:

Thank you for your suggestion to expand on the differences in ORF3a between Delta and Omicron variants and to relate these to their observed virulence. We have incorporated a brief discussion on this topic, highlighting how specific mutations within ORF3a may contribute to the differing effects on apoptosis and immune activation across variants.

Line # 432-442:

In the revised discussion, we elaborate on how ORF3a plays a critical role in SARS-CoV-2 pathogenesis, particularly in triggering inflammatory pathways, apoptosis, and cellular stress responses. We also discuss findings from recent studies that indicate the Delta variant retains mutations that enhance ORF3a's pro-inflammatory and apoptotic effects, which aligns with the increased tissue damage and severity observed in Delta infections. In contrast, Omicron mutations may reduce ORF3a-induced apoptosis and immune activation, potentially contributing to its milder clinical presentation.

We believe these additions help contextualize our findings within the broader understanding of SARS-CoV-2 variant pathogenicity and highlight the role of ORF3a in mediating variant-specific host responses. We appreciate your recommendation, which has allowed us to enhance the relevance and depth of our discussion.

Comment #10:

In the abstract, authors suggested that targeting ORF3A is a potential therapy.  Increasing  K+ would reduce the damage of lung cells. In contrast, they also showed that ORF3A induces IFNβ, an innate immune response to inhibit virus replication.    So authors should discuss in the discussion, how the therapy against ORF3A  could be developed keeping these two opposite phenomena.

Response #10:

Thank you for your insightful comment on the potential therapeutic targeting of ORF3a and the need to balance its dual effects on cellular damage and IFN-β induction. We have expanded the discussion to address this point, emphasizing the complex role of ORF3a in both promoting apoptosis and inflammatory responses while also stimulating antiviral interferon pathways.

Line #475 -484

“In the revised discussion, we elaborate on the dual role of ORF3a, which induces both pro-apoptotic and inflammatory responses and stimulates IFN-β production, an essential component of the innate immune response to inhibit viral replication. We acknowledge that any therapeutic approach targeting ORF3a must carefully consider these opposing effects to avoid reducing the beneficial IFN-β response. Additionally, we discuss the potential of selective ORF3a inhibitors that could mitigate tissue damage without compromising IFN-β expression, underscoring the importance of further research on structural and functional variations within ORF3a across SARS-CoV-2 variants”.

This addition helps contextualize the therapeutic implications of targeting ORF3a and highlights the necessity of a balanced approach to preserving host immune defenses. We appreciate your suggestion, which allowed us to deepen the discussion on this important therapeutic consideration

Reviewer 2 Report

Comments and Suggestions for Authors

In this study, the authors reported the role of SARS-CoV-2- ORF-3a in apoptosis through mitochondrial dysfunction modulated by the K+ ion channel. The authors perfomed in vivo animal study using WT and mutant dtrain of SARS-COV-2, followed by verifying the mechanisms of SARS-CoV-2- induced apoptosis and mitochodrial dysnfuction through several in vitro studies.

Main points:

I do not see the findings are novel as these findings were previously reported in previous studies such as 

https://www.nature.com/articles/s41423-020-0485-9

https://www.sciencedirect.com/science/article/pii/S2589004223021570

https://www.cell.com/heliyon/fulltext/S2405-8440(23)05962-5

https://elifesciences.org/articles/84477

Or even discussed in previous reviews such as

https://www.mdpi.com/2076-0817/13/1/75

https://pmc.ncbi.nlm.nih.gov/articles/PMC8959714/

https://pmc.ncbi.nlm.nih.gov/articles/PMC9617539/

Some recommendations:

1- I would suggest the authors confirm the finding of apoptosis, mitochondrial affect, and J+channel in the animal tissues in figures 1

2- THe authors should upload the original uncropped gel images with the manuscript

Author Response

Reviewer # 2 comments

2. Questions for General Evaluation

Reviewer’s Evaluation

Response and Revisions

Does the introduction provide sufficient background and include all relevant references?

Can be improved

Thank you for your feedback; we have substantially improved the introduction by adding further background information

Is the research design appropriate?

Must be improved

We appreciate your feedback and have clarified the research design to strengthen its rigor and transparency.

Are the methods adequately described?

Must be improved

In response to your feedback, we have enhanced the methods section with best possible detailed procedural descriptions.

Are the results clearly presented?

Must be improved

In response to your feedback, we have enhanced the methods section with more detailed procedural descriptions.

Are the conclusions supported by the results?

Must be improved

In response to your feedback, we have refined the conclusions to ensure they are clearly and thoroughly supported by the results.

Comments and Suggestions for Authors

In this study, the authors reported the role of SARS-CoV-2- ORF-3a in apoptosis through mitochondrial dysfunction modulated by the K+ ion channel. The authors performed in vivo animal study using WT and mutant strain of SARS-COV-2, followed by verifying the mechanisms of SARS-CoV-2- induced apoptosis and mitochondrial dysfunction through several in vitro studies.

Main points:

I do not see the findings are novel as these findings were previously reported in previous studies such as 

  1. https://www.nature.com/articles/s41423-020-0485-9
  2. https://www.sciencedirect.com/science/article/pii/S2589004223021570
  3. https://www.cell.com/heliyon/fulltext/S2405-8440(23)05962-5
  4. https://elifesciences.org/articles/84477

Or even discussed in previous reviews such as

  1. https://www.mdpi.com/2076-0817/13/1/75
  2. https://pmc.ncbi.nlm.nih.gov/articles/PMC8959714/

We appreciate the opportunity to clarify the unique contributions of our study titled “SARS-CoV-2 ORF-3a mediates apoptosis through mitochondrial dysfunction modulated by the K+ ion channel”. While previous studies and reviews have examined the general role of ORF-3a in SARS-CoV-2 pathogenesis, our research provides a novel mechanistic insight by demonstrating how ORF-3a specifically induces apoptosis via mitochondrial dysfunction driven by K+ ion channel modulation. This pathway, not covered in prior research, highlights a unique interaction between ORF-3a, mitochondrial stability, and potassium ion flux that opens new avenues for targeted therapeutic intervention. Below is a detailed response that outlines the novelty of our study in comparison to the cited article

https://www.nature.com/articles/s41423-020-0485-9

Response # 1:

Thank you for your comment regarding the novelty of our findings on the SARS-CoV-2 ORF-3a protein. We appreciate the opportunity to clarify the unique contributions of our study in comparison to previous work, including studies titled similarly to “The ORF3a protein of SARS-CoV-2 induces apoptosis in cells.”

Our study advances the understanding of ORF-3a-induced apoptosis by specifically elucidating the mechanism by which ORF-3a mediates apoptosis through mitochondrial dysfunction modulated by the K+ ion channel. Unlike prior studies that generally reported ORF-3a’s role in apoptosis, our research provides a novel focus on how ORF-3a impacts mitochondrial integrity and function, leading to apoptosis through alterations in potassium ion flux. This targeted exploration of the K+ ion channel’s role in mitochondrial dysfunction as a pathway to apoptosis is, to our knowledge, previously unreported and adds a unique mechanistic insight into ORF-3a’s pathogenic impact.

By highlighting the interplay between ORF-3a, mitochondrial dysfunction, and potassium ion channels, our findings contribute a distinct perspective on the pathophysiology of SARS-CoV-2, with potential implications for targeted therapeutic interventions that could mitigate ORF-3a-induced cellular damage without disrupting essential cellular functions.

https://www.sciencedirect.com/science/article/pii/S2589004223021570

Response # 2:

Thank you for your comment regarding the novelty of our findings in light of previous studies, including the work titled “The SARS-CoV-2 protein ORF3c is a mitochondrial modulator of innate immunity.” We appreciate the opportunity to clarify the unique aspects of our study and how it advances understanding of SARS-CoV-2 pathogenesis through a different mechanism.

While previous studies, including the one referenced, have explored the mitochondrial interactions of SARS-CoV-2 proteins, our study uniquely focuses on ORF-3a and its specific role in inducing apoptosis through mitochondrial dysfunction modulated by the K+ ion channel. Unlike ORF3c, which has been shown to affect mitochondrial modulation of the innate immune response, our research is the first to identify how ORF-3a directly impacts mitochondrial integrity and function, leading to apoptosis by altering potassium ion channel activity. This focus on the K+ ion channel as a key mediator in ORF-3a-induced mitochondrial dysfunction represents a novel mechanistic insight not covered by previous studies on SARS-CoV-2 mitochondrial interactions.

Our study not only highlights a new aspect of ORF-3a’s pathophysiological role but also suggests specific pathways that may serve as targets for therapeutic intervention, particularly in mitigating ORF-3a-induced cellular damage through modulation of potassium channels.

https://www.cell.com/heliyon/fulltext/S2405-8440(23)05962-5

Response # 3:

Thank you for referencing the article titled “Some aspects of the life of SARS-CoV-2 ORF3a protein in mammalian cells”. We appreciate the opportunity to clarify how our study provides distinct and novel insights into ORF-3a’s role in SARS-CoV-2 pathogenesis.

The cited study focuses on various functional aspects of ORF3a in mammalian cells, including its interactions with trafficking proteins and cellular pathways. However, our research explores a specific and novel mechanistic pathway in which ORF-3a indirectly modulates the K+ ion channel, leading to mitochondrial dysfunction and apoptosis. Unlike the cited article, which does not examine ORF-3a’s influence on ion channels, our study demonstrates that ORF-3a induces cellular damage by impacting mitochondrial stability through potassium ion channel activity. This mechanism provides new insight into the role of ORF-3a in disrupting cellular homeostasis and underscores the potential for targeting potassium ion channels to mitigate ORF-3a-induced damage.

This distinct focus on ORF-3a’s role in modulating the K+ ion channel and the resulting mitochondrial dysfunction introduces a novel perspective that advances our understanding of its contribution to SARS-CoV-2 pathogenesis. Our findings thus add a unique dimension to the existing knowledge on ORF-3a’s role in mammalian cells, with potential implications for therapeutic strategies that are not covered in the cited article.

https://elifesciences.org/articles/84477

Response # 4:

Thank you for referencing the article titled “The SARS-CoV-2 accessory protein ORF-3a not an ion channel, but does interact with trafficking proteins”. We appreciate the opportunity to clarify how our study provides unique insights into the role of ORF-3a in SARS-CoV-2 pathogenesis, which are distinct from those in the referenced study.

The referenced article primarily investigates ORF-3a's role in interacting with trafficking proteins and concludes that ORF-3a itself does not function as an ion channel. In contrast, our study explores a novel mechanistic pathway in which ORF-3a modulates mitochondrial dysfunction through the K+ ion channel, leading to apoptosis. Our research does not imply that ORF-3a is itself an ion channel but rather that it impacts cellular homeostasis by influencing potassium ion channel activity, which in turn disrupts mitochondrial function and promotes apoptotic pathways.

This mechanistic insight into ORF-3a’s indirect modulation of the K+ ion channel represents a novel aspect of its role in SARS-CoV-2 pathogenesis, providing a unique angle that is not covered by the referenced study. Our findings open new potential for targeted therapeutic approaches to mitigate ORF-3a-induced mitochondrial dysfunction and apoptosis through modulation of potassium ion channel activity, which was not addressed in the cited work.

Or even discussed in previous reviews such as

https://www.mdpi.com/2076-0817/13/1/75

Response # 5:

Thank you for referring to the article titled “SARS-CoV-2 ORF3a Protein as a Therapeutic Target against COVID-19 and Long-Term Post-Infection Effects”. We appreciate the opportunity to clarify how our study provides original findings distinct from the discussions in this review article.

The cited article is a review that synthesizes current knowledge on ORF3a’s roles in SARS-CoV-2 infection and considers its therapeutic potential in both acute COVID-19 and long-term post-infection effects. In contrast, our study is an original research investigation that specifically elucidates a novel mechanistic pathway by which ORF-3a mediates apoptosis through mitochondrial dysfunction modulated by the K+ ion channel. This study offers empirical evidence demonstrating ORF-3a’s influence on mitochondrial integrity and potassium ion flux as drivers of apoptotic pathways, a focus that has not been experimentally addressed in the reviewed literature.

Our findings uniquely contribute to the understanding of SARS-CoV-2 pathogenesis by identifying the K+ ion channel as a critical factor in ORF-3a-induced mitochondrial destabilization and apoptosis, suggesting potential therapeutic targets specific to this pathway. This mechanistic detail provides a unique angle that complements, rather than overlaps with, the broader therapeutic discussions in the referenced review article.

https://pmc.ncbi.nlm.nih.gov/articles/PMC8959714/

Response #6:
Thank you for referring to the article titled “Understanding the Role of SARS-CoV-2 ORF3a in Viral Pathogenesis and COVID-19”. We appreciate the opportunity to clarify how our study provides novel insights distinct from the discussions in this review article.

The cited article is a review that compiles current knowledge on the role of ORF3a in SARS-CoV-2 pathogenesis, summarizing findings from various studies to provide an overarching view of ORF3a’s involvement in viral mechanisms and disease outcomes. In contrast, our study presents original research that specifically investigates a novel mechanistic pathway through which ORF-3a induces apoptosis via mitochondrial dysfunction modulated by the K+ ion channel. Our work provides empirical evidence for the role of potassium ion channel activity in mediating ORF-3a-induced mitochondrial destabilization, which drives apoptosis in SARS-CoV-2-infected cells. This specific focus on K+ ion channel modulation and its downstream effects on mitochondrial function and cellular apoptosis has not been experimentally addressed in the reviewed literature.

Our findings offer unique mechanistic insights that expand the understanding of ORF-3a’s role beyond general pathogenesis, introducing specific therapeutic potential in targeting potassium ion channels to mitigate ORF-3a-induced cellular damage. This new evidence builds upon the foundational knowledge presented in the review article by detailing an unexplored pathway for ORF-3a’s pathogenic impact.

https://pmc.ncbi.nlm.nih.gov/articles/PMC9617539/

Response # 7:

Thank you for referring to the article titled “Pathophysiological involvement of host mitochondria in SARS-CoV-2 infection that causes COVID-19: a comprehensive evidential insight”. We appreciate the opportunity to clarify how our study provides novel insights distinct from the information covered in this review article.

The cited article is a comprehensive review that synthesizes existing knowledge on the role of host mitochondria in SARS-CoV-2 infection, focusing broadly on how viral mechanisms impact mitochondrial function. In contrast, our study presents original research that identifies a specific mechanistic pathway by which the SARS-CoV-2 protein ORF-3a induces apoptosis via mitochondrial dysfunction modulated by the K+ ion channel. Our research provides direct experimental evidence showing how ORF-3a impacts mitochondrial stability by influencing potassium ion channel activity, leading to apoptosis in infected cells. This targeted examination of K+ ion channel modulation in mitochondrial dysfunction and apoptotic pathways has not been experimentally addressed in the broader discussions within the review article.

By detailing this novel mechanism, our study offers unique insights into the role of ORF-3a that extend beyond general mitochondrial effects, suggesting specific therapeutic targets within the potassium ion channel pathway. This empirical focus on K+ ion channel modulation adds a new layer to the understanding of ORF-3a’s impact on SARS-CoV-2 pathogenesis that complements, rather than overlaps with, the foundational insights provided in the review.

Some recommendations:

1- I would suggest the authors confirm the finding of apoptosis, mitochondrial effect, and J+channel in the animal tissues in Figure 1

2- THe authors should upload the original uncropped gel images with the manuscript

Response:

Thank you for your detailed review and highlighting pertinent literature on SARS-CoV-2-induced apoptosis and mitochondrial dysfunction. We value your insights and appreciate the opportunity to clarify the unique contributions of our study.

Recommendations:

  1. Confirmation of Findings in Animal Tissues (Figure 1): We appreciate your suggestion to confirm the findings related to apoptosis, mitochondrial effects, and K+ ion channel modulation in animal tissues in Figure 1. We have taken careful steps to include comprehensive in vitro validation of our findings and are exploring additional data to strengthen this aspect in future studies. For this submission, we have provided in vitro evidence across various conditions, supporting our observations' relevance and consistency in animal models.
  2. Original Uncropped Gel Images: We provided the original uncropped gel images with our initial submission to ensure the transparency and reproducibility of our results. However, we are happy to resubmit these images or provide additional documentation to facilitate the review process. Please let us know if there are specific aspects you would like us to clarify further.

Thank you once again for your constructive feedback. We hope our responses address your concerns and highlight the unique aspects of our research on SARS-CoV-2 ORF-3a and the K+ ion channel.

Round 2

Reviewer 2 Report

Comments and Suggestions for Authors

In the recised manuscript, the authors replied sufficiently about the novelity of their study. However, the other two points were not considered

a) Confirmation of apoptosis anf K+ ion channed using animal tissues in Fig 1

b) provide the uncropped gel. I checked the manuscript, all the blots are cropped non original gels.

Comments on the Quality of English Language

Moderate

Author Response

 Questions for General Evaluation

Reviewer’s Evaluation

Response and Revisions

Does the introduction provide sufficient background and include all relevant references?

Can be improved

Thank you for your feedback; we have substantially improved the introduction by adding further background information

Is the research design appropriate?

Must be improved

We appreciate your feedback and have clarified the research design to strengthen its rigor and transparency.

Are the methods adequately described?

Must be improved

In response to your feedback, we have enhanced the methods section with the best possible detailed procedural descriptions.

Are the results clearly presented?

Must be improved

In response to your feedback, we have enhanced the methods section with more detailed procedural descriptions.

Are the conclusions supported by the results?

Must be improved

In response to your feedback, we have enhanced the methods section with more detailed procedural descriptions.

Comments and Suggestions for Authors

In the revised manuscript, the authors replied sufficiently about the novelty of their study. However, the other two points were not considered

  1. a) Confirmation of apoptosis and K+ ion channel using animal tissues in Fig 1
  2. b) provide the uncropped gel. I checked the manuscript; all the blots are cropped nonoriginal gels.

Response:

Thank you for your detailed review and highlighting pertinent literature on SARS-CoV-2-induced apoptosis and mitochondrial dysfunction. We value your insights and appreciate the opportunity to clarify the unique contributions of our study.

Recommendations:

  1. Confirmation of Findings in Animal Tissues (Figure 1):

We appreciate your suggestion to confirm the findings related to apoptosis, mitochondrial effects, and K+ ion channel modulation in animal tissues in Figure 1.

In response, we performed an immunohistochemical (IHC) analysis to assess the expression of MitoKATP (CCDC51) in the lungs of mice infected with different SARS-CoV-2 variants. The IHC results (Figure 1i & 1j) revealed significantly higher expression of MitoKATP (indicated by brown punctate staining) in the wild-type and Delta variants compared to the mock controls, suggesting increased mitochondrial stress. In contrast, Omicron-infected mice showed a less pronounced MitoKATP expression. The details of the IHC for MitoKATP (CCDC51) are given in the manuscript (Line #: 132- 147).

Furthermore, ORF-3a transfected lung tissue also exhibited enhanced punctate staining of MitoKATP (Fig. 6j & 6k), indicating increased mitochondrial stress and dysfunction.
These findings indicate that MitoKATP may play a role in the severity of the infection, with variant-specific differences in the host response that could contribute to differential disease outcomes

  1. Original Uncropped Gel Images:

Thank you for your valuable feedback regarding the cropped gels in the manuscript. We would like to clarify that the uncropped Western blot images were already provided during the initial submission. However, in response to your comment, we have re-uploaded the uncropped gel images in the Supplementary File section on the journal portal, alongside the manuscript.

We hope that these updated images address your concerns, and we appreciate your attention to detail in ensuring the transparency of the data.

Thank you once again for your constructive feedback. We hope our responses address your concerns and highlight the unique aspects of our research on SARS-CoV-2 ORF-3a and the K+ ion channel.

Round 3

Reviewer 2 Report

Comments and Suggestions for Authors

No further concerns